# Associative learning and extinction of conditioned threat predictors across sensory modalities

Laura. R. Koenen[1], Robert. J. Pawlik[2], Adriane Icenhour[3], Liubov Petrakova[2], Katarina Forkmann[3], Nina Theysohn[4], Harald Engler [1] & Sigrid Elsenbruch [2,3] ✉

The formation and persistence of negative pain-related expectations by classical conditioning remain incompletely understood. We elucidated behavioural and neural correlates involved in the acquisition and extinction of negative expectations towards different threats across sensory modalities. In two complementary functional magnetic resonance imaging studies in healthy humans, differential conditioning paradigms combined interoceptive visceral pain with somatic pain (study 1) and aversive tone (study 2) as exteroceptive threats. Conditioned responses to interoceptive threat predictors were enhanced in both studies, consistently involving the insula and cingulate cortex. Interoceptive threats had a greater impact on extinction efficacy, resulting in disruption of ongoing extinction (study 1), and selective resurgence of interoceptive CS-US associations after complete extinction (study 2). In the face of multiple threats, we preferentially learn, store, and remember interoceptive danger signals. As key mediators of nocebo effects, conditioned responses may be particularly relevant to clinical conditions involving disturbed interoception and chronic visceral pain.

[1] Institute of Medical Psychology and Behavioral Immunobiology, Center for Translational Neuro- and Behavioral Sciences, University Hospital Essen, University of Duisburg-Essen, Essen, Germany. [2] Department of Medical Psychology and Medical Sociology, Faculty of Medicine, Ruhr University Bochum, Bochum, Germany. [3] Translational Pain Research Unit, Department of Neurology, University Hospital Essen, University of Duisburg-Essen, Essen, Germany. [4] Institute of Diagnostic and Interventional Radiology and Neuroradiology, University Hospital Essen, University of Duisburg-Essen, Essen, Germany. ✉email: sigrid.elsenbruch@ruhr-uni-bochum.de

Expectations shape our experience of reality, and are essential for adaptive behaviour in any complex environment. In the face of danger, negative expectations are formed and dynamically updated by experience, involving associative learning and memory processes. As a key emotional response during the expectation of threat, conditioned fear is essential to trigger adaptive escape or avoidance responses. Learned fear can however also turn maladaptive and contribute to pathology, as underscored by knowledge from fear conditioning accomplished in the context of anxiety, psychological trauma, and stress-related disorders[1]. More recently, the scope has been broadened to acute and chronic pain[2], embedded within the fear-avoidance model[3]. In keeping with its biological salience, pain is a ubiquitous and fundamentally threatening experience. Interoceptive pain arising from visceral organs appears to be particularly threatening and fear-inducing[4,5], resulting in much suffering in highly prevalent disorders of the gut–brain axis like the irritable bowel syndrome (IBS)[6]. Interoceptive, visceral pain is highly modifiable by cognitions and emotions[6,7], including negative expectations during the anticipation of pain as key mediators of nocebo effects[8–12]. Despite broad clinical implications of nocebo effects reaching far beyond chronic pain[13–15], the formation and persistence of negative pain-related expectations by classical conditioning remain incompletely understood, especially with respect to neurobiological mechanisms and their possible specificity to threat modality.

Human fear conditioning studies with experimental pain as unconditioned stimuli (US) have implemented exteroceptive, somatic[16–19] or interoceptive, visceral pain as salient threats[20–23], but knowledge about common and distinct threat-specific neural mechanisms remains limited, especially regarding the extinction and retrieval of pain-related fear memories[24,25]. Combining interoceptive and exteroceptive threats from different sensory modalities, as accomplished herein, constitutes a unique opportunity to elucidate specificity to threat modality in a clinically-relevant context[4,26], and is timely given recent conceptual advances regarding interoception[27,28] and interoceptive psychopathology[29]. The experience of multiple threats from different sensory modalities closely mimics the clinical reality of patients with diverse symptoms, especially those with complex comorbidities as they often characterise patients with chronic pain. In fact, any normal environment presents multiple salient threats, which may impact learning and memory processes relevant to nocebo effects that remain incompletely understood even in healthy individuals. Existing studies with multiple threats support the role of the insula, a key region of the salience network, in sensory modality-specific effects underlying aversive expectancy[26,30–32]. The engagement of the salience network, together with regions of the fear and extinction networks, remains to be tested not only for the formation but especially for the extinction of conditioned responses to multiple threats. Conditioned negative expectations may be markedly resistant to extinction, as suggested by studies involving somatic pain stimuli[33,34]. This may be particularly the case for interoceptive memory traces, as suggested by early classical interoceptive conditioning studies carried out by soviet psychologists[35], complemented by modern approaches on fear learning of interoceptive and exteroceptive cues[36] and on the partially distinct neural representation of aversive visceral signals[37]. Impaired extinction efficacy and other phenomena related to memory processes can reportedly facilitate the return of fear and increase the risk of relapse[38], with broad implications for the chronicity and treatment of pain and fear-related disorders[39].

We herein elucidated the behavioural and neural mechanisms involved in the acquisition and extinction of negative expectations towards different types of interoceptive and exteroceptive threats across sensory modalities. To this end, we analysed data from two independent differential fear conditioning studies with methodological and conceptual overlap, allowing to assess reproducibility, and offering converging insight into conditioned anticipatory responses to threats from different sensory modalities. In both functional magnetic resonance imaging (fMRI) studies, visceral pain induced by rectal distension was implemented as clinically-relevant interoceptive US together with an exteroceptive US, which was either an equally painful thermal stimulus (study 1), or a non-nociceptive, yet equally aversive tone (study 2). In addition to threat modality-specific predictive cues (conditioned stimuli, CS), unpaired safety cues in both studies allowed us to compare conditioned differential responses to interoceptive versus exteroceptive threat predictors for different phases of conditioning. For the acquisition, we tested the general hypothesis that in the face of multiple threats, predictive learning is shaped by the salience of the US, as suggested by preparedness theory[40] and the evolutionary significance of the interoceptive modality, as illustrated by one-trial learning phenomena like conditioned nausea and taste aversion[41]. Given initial evidence that pain-modality shapes not only the perception and processing of stimuli[4,5,42], but also anticipatory responses including conditioned fear[26,36], we expected greater differential conditioned responses involving regions of the fear and salience networks to cues predicting interoceptive threat. We further tested whether conditioned responses to interoceptive threat predictors are more resistant to effective extinction, involving regions of the extinction network. The return of conditioned responses induced by reinstatement, i.e. unexpected re-exposure to the US, constitutes a promising translational tool to assess extinction efficacy[38] that has rarely been applied in brain imaging studies on pain-related fear[20]. To this end, our paradigms incorporated different reinstatement procedures following extinction phases, allowing us to test in reinstatement-test phases if the interoceptive CS–US association is more susceptible to reinstatement effects.

In sum, conditioned responses to interoceptive threat predictors were enhanced in both studies after the acquisition, consistently involving the insula and cingulate cortex as key regions of the salience network. Our results supported that unexpected exposure to interoceptive threats had a greater impact on extinction efficacy, resulting in disruption of ongoing extinction (study 1), and selective resurgence of interoceptive CS–US associations after complete extinction (study 2). Together, our findings are an important step towards unravelling how negative expectations are shaped by associative learning and memory processes in the face of multiple threats. A more refined understanding of conditioned nocebo effects in the context of clinically-relevant interoceptive and exteroceptive threats may ultimately contribute to an improved consideration of expectancy effects to the benefit of patients, broaden the rapidly evolving scope of the gut–brain axis in health neuroscience and disease[43–45], and fits into a framework of the cognitive neurosciences interfacing between mind and body[27].

## Results

**Participants**. Out of a total of $N = 77$ healthy adults who participated, $N = 12$ were excluded due to technical difficulties with MRI data acquisition ($N = 6$), movement artefacts ($N = 4$), or failure to reach visceral pain threshold within predetermined maximal distension pressure ($N = 2$). As a result, we herein report on data from $N = 42$ volunteers for study 1 (all female, age $34.5 \pm 2.0$ years; BMI $22.7 \pm 0.4$ kg/m$^2$), and $N = 23$ volunteers for study 2 (10 female, age $26.7 \pm 1.0$ years; BMI $22.4 \pm 0.7$ kg/m$^2$). Consistent with stringent and highly-parallelised exclusion

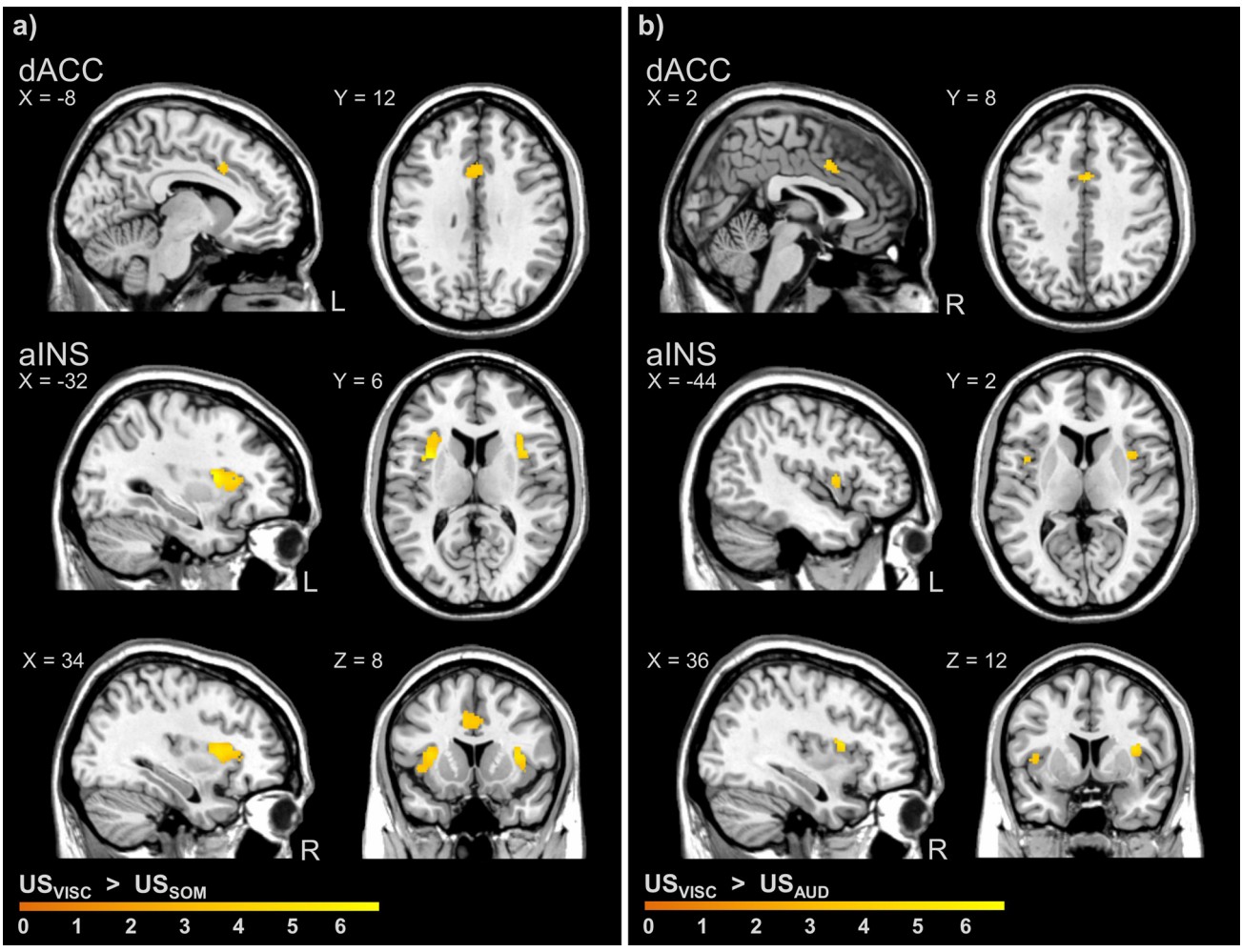

**Fig. 1 Differences in neural activation induced by interoceptive versus exteroceptive threats implemented as unconditioned stimuli (US) during acquisition in studies 1 and 2.** Interoceptive threat ($US_{VISC}$) induced significantly greater neural activation in dACC and aINS compared to both exteroceptive threats (**a** study 1, compared to $US_{SOM}$; **b** study 2, compared to $US_{AUD}$; all $P_{FWE} < 0.05$). Moreover, interoceptive and exteroceptive threats induced shared neural activation in different brain regions across studies (full results in Supplementary Table S3). Neural activations in regions of interest were superimposed on a structural T1-image and thresholded at $P < 0.001$ uncorrected for visualisation purposes; colour bars indicate $t$-scores. For details, see Table 1. For whole-brain results on differential activation, see Supplementary Fig. S4. aINS anterior insula, AUD auditory, dACC dorsal anterior cingulate cortex, FWE family-wise error, SOM somatic, US unconditioned stimuli, VISC visceral.

criteria, in both samples mean gastrointestinal symptom scores (study 1: 3.73 ± 0.4; study 2: 1.65 ± 0.4), and HADS scores for anxiety (study 1: 4.24 ± 0.4; study 2: 3.00 ± 0.4) and depression (study 1: 2.61 ± 0.3; study 2: 1.04 ± 0.3) were low. Chronic stress scores were well-within the normal range (study 1: 36.93 ± 1.5; study 2: 33.57 ± 1.5). Pain thresholds for visceral pain (study 1: 40.0 ± 1.7 mmHg; study 2: 40.2 ± 1.6 mmHg) and for thermal heat pain (study 1: 44.9 ± 0.5 °C) were well-within the range of findings in our previous work[4,21,46]. Unpleasantness thresholds for auditory stimuli (93.96 ± 1.52 dB) SPL (range: 75–108 dB SPL) were within expected ranges and comparable to previous studies from our own group[47,48].

**Unconditioned stimuli: acquisition phase**. Negative valence was consistently greater for interoceptive compared to exteroceptive threats implemented as US in both studies, despite careful matching to US pain intensity (study 1) and US unpleasantness (study 2) prior to acquisition. This was indicated by significantly greater post-acquisition $US_{VISC}$ unpleasantness ratings compared to $US_{SOM}$ in study 1 ($US_{VISC}$: 66.14 ± 3.8 mm, $US_{SOM}$: 24.62 ± 5.1 mm; $t(41) = 6.62$; $P < 0.001$; $d = 1.43$), as well as to $US_{AUD}$ in

study 2 ($US_{VISC}$: 79.30 ± 3.2 mm, $US_{AUD}$: 62.78 ± 5.3 mm; $t(22) = 3.42$; $P = 0.005$; $d = 0.75$). Note that in study 1, post-acquisition $US_{VISC}$ were also perceived as more intense ($US_{VISC}$: 77.3 ± 1.5 mm, $US_{SOM}$: 66.2 ± 2.4 mm; $t(41) = 4.23$; $P < 0.001$; $d = 0.83$), which was highly intercorrelated with US unpleasantness within both modalities ($US_{VISC}$: $r = 0.71$; $P ≤ 0.001$; $US_{SOM}$: $r = 0.65$; $P ≤ 0.001$). At the neural level, $US_{VISC}$ induced enhanced activation when compared to $US_{SOM}$ (Fig. 1a) as well as compared to $US_{AUD}$ (Fig. 1b), involving aINS and dACC in both studies, and additionally the amygdala in study 1 (Table 1). Further, interoceptive $US_{VISC}$ consistently induced lower neural activation compared to both exteroceptive $US_{SOM}$ and $US_{AUD}$ within pINS (Table 1). Of note, these differences between US modalities remained largely unchanged when considering post-acquisition differences in US unpleasantness (study 1, 2) or US intensity ratings (study 1 only) as covariates of no interest (Supplementary Tables S1 and S2). Moreover, conjunction analyses (against global null) revealed shared neural activation in study 1 ($US_{VISC} ∩ US_{SOM}$) in the aINS, whereas shared activation was detectable in study 2 only in uncorrected whole-brain analyses but not in FWE-corrected ROI analyses ($US_{VISC} ∩ US_{AUD}$, see Supplementary Table S3).

**Table 1 Differences in neural activation induced by interoceptive versus exteroceptive threats as unconditioned stimuli (US) during acquisition.**

| Contrast | Region | H | x | y | z | t-value | P |
|---|---|---|---|---|---|---|---|
| | | | **MNI-Coordinates** | | | | |
| **Study 1 (N = 42)** | | | | | | | |
| $US_{VISC} > US_{SOM}$ | | | | | | | |
| ROI analyses | aINS | L | −32 | 6 | 12 | 6.72 | <0.001 |
| | | R | 34 | 14 | 8 | 5.46 | <0.001 |
| | dACC | L | −8 | 12 | 38 | 4.65 | 0.001 |
| | Amygdala | L | −24 | −4 | −12 | 3.42 | 0.019 |
| Whole-brain analyses | PCC | R | 4 | −40 | 12 | 3.86 | <0.001 |
| | Inferior frontal gyrus, opercular (dlPFC) | R | 38 | 8 | 32 | 3.84 | <0.001 |
| | Supramarginal gyrus (S2) | L | −62 | −26 | 28 | 4.02 | <0.001 |
| | | R | 66 | −28 | 28 | 3.63 | <0.001 |
| | Vermis | − | 0 | −54 | −36 | 5.67 | <0.001 |
| | Cerebellum | R | 32 | −52 | −28 | 4.01 | <0.001 |
| $US_{VISC} < US_{SOM}$ | | | | | | | |
| ROI analyses | pINS | R | 38 | −12 | 18 | 6.20 | <0.001 |
| Whole-brain analyses | Superior frontal gyrus, medial (dmPFC) | L | −10 | 48 | 34 | 3.72 | <0.001 |
| | PHIP | R | 20 | −6 | −22 | 4.10 | <0.001 |
| | Postcentral gyrus (S1) | L | −54 | −12 | 48 | 4.08 | <0.001 |
| | | R | 44 | −22 | 54 | 3.47 | 0.001 |
| | Postcentral gyrus (S2) | L | −62 | −6 | 30 | 3.45 | <0.001 |
| | | R | 64 | −4 | 24 | 5.41 | <0.001 |
| | SMA | R | 8 | −20 | 68 | 5.40 | <0.001 |
| | Fusiform gyrus | R | 36 | −44 | −10 | 4.41 | <0.001 |
| | Rolandic operculum | L | −44 | −14 | 18 | 4.18 | <0.001 |
| | Middle temporal gyrus | L | −62 | −8 | −14 | 5.07 | <0.001 |
| | | R | 66 | −10 | −16 | 3.42 | 0.001 |
| **Study 2 (N = 23)** | | | | | | | |
| $US_{VISC} > US_{AUD}$ | | | | | | | |
| ROI analyses | aINS | L | −44 | 2 | 8 | 4.78 | 0.004 |
| | | R | 36 | 8 | 12 | 5.71 | 0.001 |
| | dACC | R | 2 | 8 | 40 | 5.16 | 0.002 |
| Whole-brain analyses | SMA | R | 6 | 4 | 46 | 6.25 | <0.001 |
| | Rolandic operculum | L | −46 | 2 | 8 | 5.36 | <0.001 |
| $US_{VISC} < US_{AUD}$ | | | | | | | |
| ROI analyses | pINS | L | −38 | −12 | 18 | 5.22 | 0.004 |
| | | R | 38 | −10 | 16 | 5.62 | 0.002 |
| Whole-brain analyses | Middle temporal gyrus | R | 56 | −36 | 6 | 9.87 | <0.001 |
| | Superior temporal gyrus | L | −54 | −36 | 10 | 7.60 | <0.001 |
| | Rolandic operculum | R | 44 | −10 | 22 | 6.50 | <0.001 |
| | Lingual area | L | −16 | −48 | −8 | 4.31 | <0.001 |
| | | R | 20 | −52 | −10 | 5.07 | <0.001 |
| | Angular gyrus | L | −48 | −54 | 24 | 4.57 | <0.001 |
| | Precuneus | R | 6 | −52 | 44 | 4.14 | <0.001 |
| | Middle occipital gyrus | L | −44 | −76 | 8 | 4.10 | <0.001 |

Differential neural activation induced by interoceptive threat ($US_{VISC}$) compared to exteroceptive threat (study 1, $US_{SOM}$; study 2, $US_{AUD}$) implemented as unconditioned stimuli (US) during acquisition. Results of second-level paired t-tests are presented. Peak voxel indicate results of ROI analyses (cluster size $k_E \geq 3$; all $P_{FWE} < 0.05$) and whole-brain analyses (*in italic font*; cluster size $k_E \geq 10$; all $P_{uncorrected} < 0.001$). Exact unilateral P-values are provided. For a visualisation, see Fig. 1 and Supplementary Fig. S4. For analysis of shared responses, see Supplementary Table S3. For analyses controlling for US ratings, see Supplementary Tables S1 and S2.
*aINS* anterior insula, *dACC* dorsal anterior cingulate cortex, *FWE* family-wise error, *H* hemisphere, *MNI* Montreal Neurological Institute, *PCC* posterior cingulate gyrus, *pINS* posterior insula, *PHIP* parahippocampus, *S1* primary somatosensory cortex, *S2* secondary somatosensory cortex, *SMA* supplementary motor area, *US* unconditioned stimuli.

**Conditioned stimuli: acquisition phase.** Repeated CS+-US pairings during acquisition resulted in the conditioned negative valence of all threat predictors in both studies, as evidenced by significant rmANOVA time effects (Supplementary Tables S4 and S5). Interestingly, this increase in negative valence was consistently enhanced for interoceptive ($\Delta CS^+_{VISC}$) versus exteroceptive threat predictors ($\Delta CS^+_{SOM}$ and $\Delta CS^+_{AUD}$, respectively), despite comparable contingency awareness between modalities (Table 2). This indication of enhanced conditioned behavioural responses to interoceptive threat predictors was supported by significant time × modality rmANOVA interaction effects in both studies (study 1: $F_{(1,41)} = 16.71$; $P < 0.001$; $\eta_p^2 = 0.29$, Supplementary Table S4; study 2: $F_{(1,22)} = 8.19$; $P = 0.009$; $\eta_p^2 = 0.27$, Supplementary Table S5), and between-modality comparisons revealing significantly enhanced conditioned negative valence post-acquisition for $\Delta CS^+_{VISC}$ (versus $\Delta CS^+_{SOM}$: $t(41) = 4.19$; $P < 0.001$; $d = 0.59$; Fig. 2a and Supplementary Table S6; versus $\Delta CS^+_{AUD}$: $t(22) = 3.28$; $P = 0.007$; $d = 0.37$; Fig. 2b and Supplementary Table S7; as well as by enhanced SCR for $CS^+_{VISC}$ in a subset of participants in study 2, see Supplementary Fig. S1). At the neural level, threat predictors induced shared differential activation in highly overlapping brain regions across studies, including aINS, hippocampus, MCC, dACC as revealed by conjunction analysis (in study 1: $\Delta CS^+_{VISC} \cap \Delta CS^+_{SOM}$; in study 2: $\Delta CS^+_{VISC} \cap \Delta CS^+_{AUD}$; full results in Supplementary Table S8). Interestingly, differential neural responses in pINS and MCC were enhanced for $\Delta CS^+_{VISC}$

**Table 2 Contingency awareness across learning phases.**

| Study 1 (N = 42) | CS+VISC | CS+SOM | CS− | P* |
|---|---|---|---|---|
| ACQ | 77.3 ± 3.0 | 70.2 ± 3.8 | 32.7 ± 4.4 | 0.052 |
| EXT | 24.2 ± 5.0 | 25.0 ± 5.1 | 10.5 ± 2.7 | 0.767 |
| RST-TEST | | | | |
| USVISC-subgroup (N = 22) | 16.6 ± 6.8 | 10.0 ± 5.2 | 8.0 ± 3.5 | 0.680 |
| USSOM-subgroup (N = 20) | 15.9 ± 6.1 | 17.3 ± 6.7 | 16.3 ± 5.3 | 0.875 |
| | | | | |
| Study 2 (N = 23) | CS+VISC | CS+AUD | CS− | P* |
| ACQ | 80.9 ± 6.0 | 85.1 ± 4.3 | 15.5 ± 4.3 | 0.552 |
| EXT | 0.0 ± 0.0 | 2.7 ± 2.3 | 1.6 ± 1.6 | 0.237 |
| RST-TEST | 2.6 ± 1.8 | 1.7 ± 1.4 | 1.3 ± 1.1 | 0.504 |

Contingency awareness for threat-predictive conditioned stimuli (CS+) and safety cues (CS−) assessed with visual analogue scales (0–100 VAS, mm, representing % probability that a CS is followed by the specific US) after acquisition (ACQ), extinction (EXT), and reinstatement-test (RST-TEST) in study 1 (CS+VISC, CS+SOM) and study 2 (CS+VISC, CS+AUD). Note that actual CS–US contingencies for all CS+ during ACQ were 80% in study 1 and 83% in study 2; all CS were presented without US in EXT and RST-TEST phases. CS− were never paired with US in any phase. Data are given as mean ± standard error of the mean.
AUD auditory, CS conditioned stimuli, SOM somatic, US unconditioned stimuli, VISC visceral.
*Exact P-values for between-modality paired t-tests assessing differences between CS+ within each study, shown uncorrected for multiple testing.

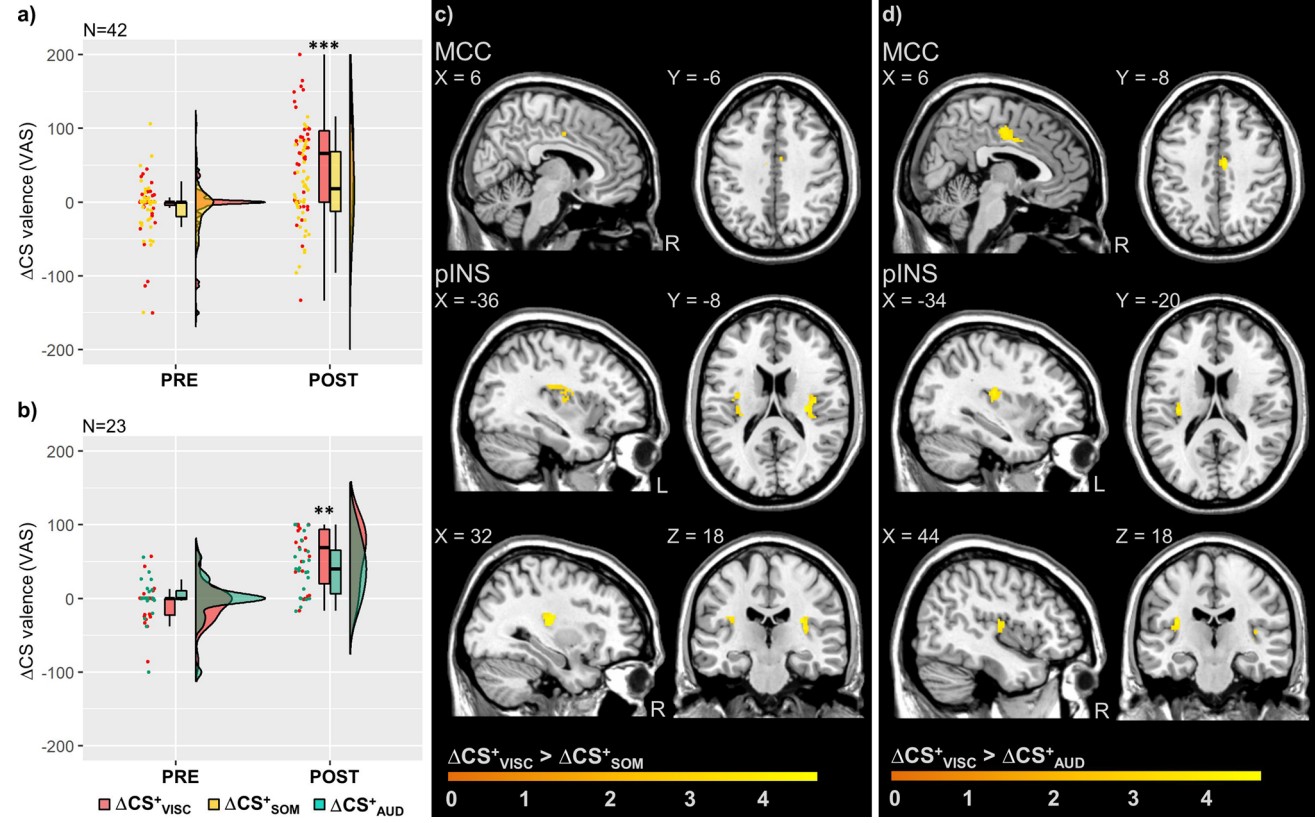

**Fig. 2 Behavioural and neural responses to cues (CS) predicting interoceptive versus exteroceptive threats during acquisition in studies 1 and 2.** Conditioned predictors of interoceptive threat (ΔCS+VISC) acquired significantly greater negative valence than conditioned predictors of exteroceptive threats after acquisition **a** study 1, compared to ΔCS+SOM: ***P < 0.001; **b** study 2, compared to ΔCS+AUD: **P < 0·01; results of Bonferroni-corrected paired t-tests between modalities, full details in Supplementary Tables S4–S7. Individual delta (Δ) scores were computed for differential CS valence of each CS+ relative to the CS−. Data are presented as individual data points, boxplots, and densities (raincloud plots[117]). At the neural level, interoceptive threat predictors (ΔCS+VISC) induced enhanced differential neural responses in MCC and pINS compared to both exteroceptive threat predictors **c** study 1, ΔCS+VISC > ΔCS+SOM; **d** study 2, ΔCS+VISC > ΔCS+AUD; all $P_{FWE}$ < 0.05; details in Table 3, but threat predictors also induced shared differential activation in overlapping brain regions across studies (full results in Supplementary Table S8). For whole-brain results, see Supplementary Fig. S5. Neural activations in regions of interest were superimposed on a structural T1-image and thresholded at P < 0.001 uncorrected for visualisation purposes; colour bars indicate t-scores. AUD auditory, CS conditioned stimuli, FWE family-wise error, MCC midcingulate cortex, pINS posterior insula, SOM somatic, VAS visual analogue scale, VISC visceral.

**Table 3 Differences in neural activation induced by cues (CS) predicting interoceptive versus exteroceptive threats during acquisition.**

| Contrast | Region | MNI-coordinates | | | | t-value | P |
|---|---|---|---|---|---|---|---|
| | | H | x | y | z | | |
| Study 1 (N = 42) | | | | | | | |
| $\Delta CS^+_{VISC} > \Delta CS^+_{SOM}$ | | | | | | | |
| ROI analyses | pINS | L | −36 | −8 | 18 | 4.31 | 0.008 |
| | | R | 32 | −22 | 12 | 4.89 | 0.002 |
| | MCC | R | 6 | −6 | 40 | 3.56 | 0.023 |
| Whole-brain analyses | Superior frontal gyrus, medial (vmPFC) | R | *12* | *60* | *0* | *3.62* | *<0.001* |
| | Precentral gyrus (S1) | R | *14* | *−30* | *74* | *5.41* | *<0.001* |
| | Rolandic operculum | L | *−42* | *−30* | *16* | *3.49* | *0.001* |
| | Thalamus | L | *−16* | *−26* | *0* | *3.79* | *<0.001* |
| | | R | *14* | *−22* | *0* | *4.14* | *<0.001* |
| | Putamen | R | *28* | *−8* | *14* | *5.25* | *<0.001* |
| | Inferior temporal gyrus | L | *−58* | *−50* | *−14* | *3.66* | *<0.001* |
| | Middle occipital lobe | L | *−32* | *−80* | *40* | *4.17* | *<0.001* |
| | Vermis | R | *4* | *−44* | *−16* | *4.12* | *<0.001* |
| | Cerebellum | L | *−36* | *−74* | *−34* | *4.29* | *<0.001* |
| | | R | *36* | *−78* | *−26* | *4.84* | *<0.001* |
| $\Delta CS^+_{VISC} < \Delta CS^+_{SOM}$ | – | – | – | – | – | – | – |
| Study 2 (N = 23) | | | | | | | |
| $\Delta CS^+_{VISC} > \Delta CS^+_{AUD}$ | | | | | | | |
| ROI analyses | pINS | L | −34 | −20 | 18 | 4.86 | 0.007 |
| | | R | 44 | −14 | 14 | 4.39 | 0.017 |
| | MCC | R | 6 | −10 | 42 | 4.77 | 0.004 |
| | dACC | R | 4 | 6 | 36 | 3.80 | 0.026 |
| Whole-brain analyses | Superior temporal gyrus | L | *−42* | *−28* | *8* | *6.05* | *<0.001* |
| | | R | *50* | *−22* | *8* | *6.72* | *<0.001* |
| $\Delta CS^+_{VISC} < \Delta CS^+_{AUD}$ | – | – | – | – | – | – | – |

Differential neural activation induced by predictors of interoceptive (CS+VISC) versus exteroceptive threat (study 1, CS+SOM; study 2, CS+AUD) relative to safety-predictive CS−, during acquisition. Results of second-level paired *t*-tests (study 1: {CS+VISC < CS−} > {CS+SOM < CS−}; study 2: {CS+VISC < CS−} > {CS+AUD < CS−}; and vice versa) are presented. Peak voxel indicates results of ROI analyses (cluster size $k_E \geq 3$; all $P_{FWE} < 0.05$) and whole-brain analyses (*in italic font*; cluster size $k_E \geq 10$; all $P_{uncorrected} < 0.001$). Exact unilateral *P*-values are provided. For a visualisation, see Fig. 2 and Supplementary Fig. S5. For analysis of shared responses, see Supplementary Table S8. For analyses controlling for US ratings, see Supplementary Tables S9 and S10.
*AUD* auditory, *CS* conditioned stimuli, *dACC* dorsal anterior cingulate cortex, *FWE* family-wise error, *H* hemisphere, *MCC* midcingulate cortex, *MNI* Montreal Neurological Institute, *pINS* posterior insula, *S1* primary somatosensory cortex, *SOM* somatic, *VISC* visceral, *vmPFC* ventromedial prefrontal cortex.

compared to both $\Delta CS^+_{SOM}$ (Fig. 2c), as well as compared to $\Delta CS^+_{AUD}$ (Fig. 2d), a finding that was strikingly similar between both studies ($P_{FWE} < 0.05$, Table 3). Of note, these findings were not appreciably altered when considering post-acquisition differences in US ratings as covariates of no interest (Supplementary Tables S9 and S10). Moreover, exploratory correlational analyses revealed significant correlations between CS and US valence, as well as between differential CS- and US-induced activations in peak activations in the insula and cingulate cortices (Supplementary analyses).

**Conditioned stimuli: extinction phase.** Repeated presentations of all CS without any US-induced extinction of conditioned negative valence of threat predictors (for significant rmANOVA time effects observed in both studies, see Supplementary Tables S4 and S5). However, this was shaped by CS type, as supported by significant time × modality interaction effects (study 1: $F(1,41) = 5.51$; $P = 0.024$; $\eta_p^2 = 0.12$; study 2: $F(1,22) = 6.36$; $P = 0.019$; $\eta_p^2 = 0.22$). In study 1 with an immediate extinction and a small number of extinction trials, post-extinction negative valence remained significantly greater for interoceptive compared to exteroceptive threat predictors ($\Delta CS^+_{VISC}$: 35.38 ± 7.2 mm, $\Delta CS^+_{SOM}$: 9.38 ± 7.0 mm; $t(41) = 3.42$; $P = 0.003$; $d = 0.57$). Contingency ratings indicated no differences between modalities, but an overestimation of true contingencies ($CS^+_{VISC}$: 24.2 ± 5.0%, $CS^+_{SOM}$: 25.0 ± 5.1%, with de facto 0% during extinction; Table 2). On the other hand, in study 2 with extinction accomplished 24 h after acquisition and a larger number of

extinction trials, no mid- or post-extinction difference in negative valence between interoceptive and exteroceptive cues was detectable ($\Delta CS^+_{VISC}$: 5.00 ± 4.0 mm, $\Delta CS^+_{AUD}$: 3.00 ± 2.8 mm; $P > 0.99$; $d = 0.12$; for mid-extinction results, see Supplementary Table S7), and contingency ratings were widely accurate, with no differences between modalities (Table 2). At the neural level, we observed shared differential neural activation induced by interoceptive and exteroceptive threat predictors (i.e. $\Delta CS^+_{VISC} \cap \Delta CS^+_{SOM}$ and $\Delta CS^+_{VISC} \cap \Delta CS^+_{AUD}$, respectively; full results in Supplementary Table S11) within several ROIs, including hippocampus in both studies, as well as vmPFC, pINS, and amygdala (study 1, Supplementary Fig. S2a) and aINS, and dACC (study 2, Supplementary Fig. S2b). No differences between modalities in differential CS-induced neural activation were found in any ROI in either study. However, whole-brain analyses revealed differences in the temporal gyrus (Supplementary Table S12).

**Conditioned stimuli: reinstatement-test phase.** In the reinstatement-test phase, unpaired CS presentations were implemented immediately after single threat reinstatement in study 1 [i.e. unexpected exposure to only US_VISC in one subgroup (N = 22), and to only US_SOM in another subgroup (N = 20)], or after multiple threat reinstatement (i.e. unexpected exposure to both US_VISC and US_SOM in all participants) in study 2. After single threat reinstatement (US_VISC (N = 22); US_SOM (N = 20)) in study 1, a significant time × modality × group interaction was observed for negative CS valence ($F(1,1,40) = 5.56$; $P = 0.023$; $\eta^2 = 0.12$, see Supplementary Table S4). After multiple threat reinstatement with

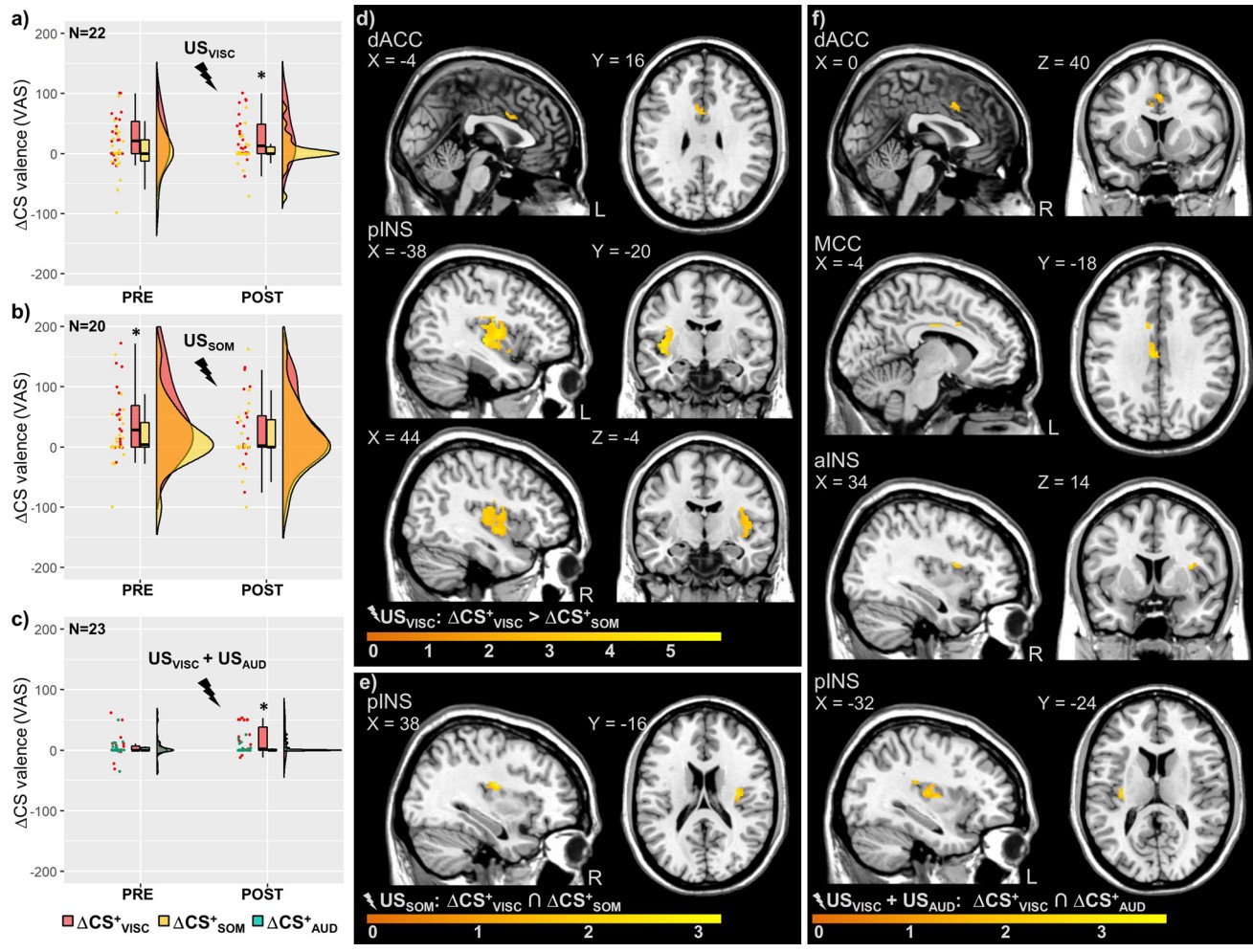

**Fig. 3 Behavioural and neural responses to cues (CS) predicting interoceptive versus exteroceptive threats during reinstatement-test in studies 1 and 2.** After reinstatement with unexpected US, negative valence was greater for interoceptive compared to exteroceptive threat predictors in reinstatement groups involving $US_{VISC}$ **a** study 1, $US_{VISC}$-subgroup, $\Delta CS^+_{VISC}$ versus $\Delta CS^+_{SOM}$; **c** study 2, $\Delta CS^+_{VISC}$ versus $\Delta CS^+_{AUD}$; *both $P < 0.05$, results of Bonferroni-corrected paired $t$-tests between modalities, full results in Supplementary Tables S4–S7, whereas no difference was observed in the $US_{SOM}$-subgroup **b** study 1, $\Delta CS^+_{VISC}$ versus $\Delta CS^+_{SOM}$). Individual delta ($\Delta$) scores were computed for differential CS valence of each $CS^+$ relative to the $CS^-$. Data are presented as individual data points, boxplots, and densities (raincloud plots[117]). At the neural level, differential activation induced by interoceptive cues was enhanced in the $US_{VISC}$-subgroup within dACC and pINS during reinstatement-test **d** study 1, $\Delta CS^+_{VISC}$ compared to $\Delta CS^+_{SOM}$, all $P_{FWE} < 0.05$). While no differential activation was observed for $\Delta CS^+_{VISC}$ compared to $\Delta CS^+_{SOM}$ in the $US_{SOM}$-subgroup in study 1 or compared to $\Delta CS^+_{AUD}$ in study 2, shared differential neural activation was induced by both threat predictors in regions of interest, such as in the insula and cingulate cortex **e** $\Delta CS^+_{VISC} \cap \Delta CS^+_{SOM}$ in study 1, $US_{SOM}$-subgroup; **f** $\Delta CS^+_{VISC} \cap \Delta CS^+_{AUD}$ in study 2. For full results, see Tables 4–6. For whole-brain results on differential activation, see Supplementary Fig. S7. Neural activations in regions of interest were superimposed on a structural T1-image and thresholded at $P < 0.01$ uncorrected for visualisation purposes; colour bars indicate $t$-scores. aINS anterior insula, AUD auditory, CS conditioned stimuli, dACC dorsal anterior cingulate cortex, FWE family-wise error, MCC midcingulate cortex, pINS posterior insula, SOM, somatic, US unconditioned stimuli, VAS visual analogue scale, VISC visceral.

both $US_{VISC}$ and $US_{SOM}$ in study 2, the interaction effect was not significant ($P = 0.055$; $\eta_p^2 = 0.16$, full results in Supplementary Table S5). Planned between-modality comparisons for the POST RST-TEST time point (Supplementary Tables S6 and S7) revealed greater differential negative valence of interoceptive compared to exteroceptive threat predictors in those groups involving $US_{VISC}$ exposure during reinstatement (study 1, $US_{VISC}$-subgroup, $\Delta CS^+_{VISC}$ versus $\Delta CS^+_{SOM}$, $t(21) = 2.73$; $P = 0.025$; $d = 0.58$; Fig. 3a; study 2, $\Delta CS^+_{VISC}$ versus $\Delta CS^+_{AUD}$, $t(22) = 2.49$; $P = 0.042$; $d = 0.73$; Fig. 3c), whereas no difference was observed in the $US_{SOM}$-subgroup (study 1, $\Delta CS^+_{VISC}$ versus $\Delta CS^+_{SOM}$, $P = 0.355$; $d = 0.21$; Fig. 3b). Exploratory within-group comparisons (PRE-POST) for study 1 showed no significant changes for $\Delta CS^+_{VISC}$ and $\Delta CS^+_{SOM}$ (both $P > 0.05$) in the $US_{VISC}$-subgroup, whereas in the $US_{SOM}$-subgroup $\Delta CS^+_{VISC}$ significantly decreased ($P = 0.005$ uncorrected, Supplementary

Table S6). The same comparisons for study 2 (PRE-POST) revealed a significant increase for $\Delta CS^+_{VISC}$ ($P = 0.038$ uncorrected), and no change for $\Delta CS^+_{AUD}$ ($P > 0.05$) (Supplementary Table S7).

At the neural level, threat predictors induced shared differential neural activation within hippocampus and vmPFC in the $US_{VISC}$-subgroup (Table 4), and in the pINS in the $US_{SOM}$-subgroup (study 1, $\Delta CS^+_{VISC} \cap \Delta CS^+_{SOM}$, Fig. 3e, Table 5). After multiple threat reinstatement in study 2, threat predictors induced shared activation in the hippocampus, aINS, pINS, dACC, and MCC ($\Delta CS^+_{VISC} \cap \Delta CS^+_{SOM}$, Fig. 3f, Table 6). Differential activation induced by interoceptive versus exteroceptive threat predictors was enhanced within pINS (L: $x = -38$, $y = -20$, $z = 0$, $t(21) = 5.23$, $P_{FWE} = 0.004$, R: $x = 44$, $y = -4$, $z = -4$, $t(21) = 4.91$, $P_{FWE} = 0.007$) and dACC (L: $x = -4$, $y = 16$, $z = 22$, $t(21) = 4.55$, $P_{FWE} = 0.023$) only in the $US_{VISC}$-subgroup (study 1, $\Delta CS^+_{VISC}$ compared to

**Table 4 Neural activation induced by cues (CS) predicting interoceptive versus exteroceptive threats during reinstatement-test after visceral single threat reinstatement in study 1.**

| Contrast | Region | H | x | y | z | t-value | P |
|---|---|---|---|---|---|---|---|
| | | | | MNI-coordinates | | | |
| Study 1: US$_{VISC}$-subgroup ('single threat reinstatement'; N = 22) | | | | | | | |
| $\Delta$CS$^{+}_{VISC}$ ∩ $\Delta$CS$^{+}_{SOM}$ | | | | | | | |
| ROI analyses[1] | HIP | L | −16 | −40 | 6 | 2.78 | 0.009 |
| | | R | 40 | −26 | −12 | 2.96 | 0.004 |
| Whole-brain analyses[1] | Superior frontal gyrus, orbital (vmPFC) | L | −16 | 56 | −12 | 3.79 | <0.001 |
| | Inferior temporal gyrus | L | −48 | −24 | −20 | 2.48 | <0.001 |
| | Middle occipital lobe | L | −22 | −82 | 14 | 2.52 | <0.001 |
| | Cerebellum | L | −28 | −78 | −34 | 3.67 | <0.001 |
| | | R[a] | 6 | −72 | −34 | 4.61 | <0.001 |
| ROI analyses[2] | vmPFC | L | −16 | 56 | −12 | 3.79 | <0.001 |
| Whole-brain analyses[2] | PCC | L | −16 | −44 | 8 | 3.98 | <0.001 |
| | Superior frontal gyrus (dmPFC) | R | 20 | 6 | 70 | 3.00 | <0.001 |
| | Middle frontal gyrus, orbital (vlPFC) | L[a] | −36 | 46 | −4 | 4.56 | <0.001 |
| | | R | 38 | 46 | −2 | 2.93 | <0.001 |
| | Inferior frontal gyrus, triangular (vlPFC) | L | −38 | 20 | 6 | 2.53 | <0.001 |
| | Superior parietal gyrus | R | 34 | −54 | 58 | 2.86 | <0.001 |
| | Inferior parietal gyrus | L | −38 | −62 | 50 | 2.40 | <0.001 |
| $\Delta$CS$^{+}_{VISC}$ > $\Delta$CS$^{+}_{SOM}$ | | | | | | | |
| ROI analyses | pINS | L | −38 | −20 | 0 | 5.23 | 0.004 |
| | | R | 44 | −4 | −4 | 4.91 | 0.007 |
| | dACC | L | −4 | 16 | 22 | 4.55 | 0.023 |
| Whole-brain analyses | MCC | R | 12 | −10 | 50 | 5.18 | <0.001 |
| | PHIP | R | 16 | −38 | −10 | 4.10 | <0.001 |
| | Thalamus | R | 20 | −24 | −2 | 3.97 | <0.001 |
| | Lingual | L | −14 | −42 | −10 | 5.32 | <0.001 |
| | | R | 14 | −52 | 6 | 4.96 | <0.001 |
| | Heschl gyrus | L | −52 | −14 | 8 | 5.44 | <0.001 |
| | Precentral gyrus | L | −36 | −28 | 66 | 4.23 | <0.001 |
| | | R | 52 | 4 | 18 | 4.42 | <0.001 |
| | Postcentral gyrus | L | −50 | −20 | 54 | 4.80 | <0.001 |
| | | R | 60 | −16 | 30 | 4.16 | <0.001 |
| | Rolandic operculum | R | 46 | −10 | 14 | 4.02 | <0.001 |
| | Precuneus | L | −16 | −58 | 62 | 5.03 | <0.001 |
| | | R | 12 | −62 | 68 | 4.08 | <0.001 |
| | Cuneus | R | 22 | −66 | 30 | 3.76 | 0.001 |
| | Superior parietal gyrus | R | 22 | −64 | 64 | 3.77 | 0.001 |
| | Calcarine fissure | L | −10 | −70 | 20 | 4.04 | <0.001 |
| | Cerebellum | R | 10 | −64 | −48 | 4.17 | <0.001 |
| $\Delta$CS$^{+}_{VISC}$ < $\Delta$CS$^{+}_{SOM}$ | – | – | – | – | – | – | – |

Shared and differential neural activation induced by predictors of interoceptive (CS$^{+}_{VISC}$) and exteroceptive threat (CS$^{+}_{SOM}$) relative to safety-predictive CS$^{−}$, during reinstatement-test (RST-TEST) after single threat reinstatement with US$_{VISC}$. For shared activation, results of conjunction analyses against global null are presented ([1]{CS$^{+}_{VISC}$ > CS$^{−}$} ∩ {CS$^{+}_{SOM}$ > CS$^{−}$}; [2]{CS$^{+}_{VISC}$ < CS$^{−}$} ∩ {CS$^{+}_{SOM}$ < CS$^{−}$}). For differential activation, results of second-level paired *t*-tests are presented ({CS$^{+}_{VISC}$ < CS$^{−}$} > {CS$^{+}_{SOM}$ < CS$^{−}$}; and vice versa). Peak voxel indicates results of ROI analyses (cluster size $k_E \geq 3$; all $P_{FWE}$ < 0.05) and whole-brain analyses (*in italic font*; cluster size $k_E \geq 10$; all $P_{uncorrected}$ < 0.001). Exact unilateral *P*-values are provided. For a visualisation, see Fig. 3 and Supplementary Fig. S7.
*CS* conditioned stimuli, *FWE* family-wise error, *dACC* dorsal anterior cingulate cortex, *dmPFC* dorsomedial prefrontal cortex, *H* hemisphere, *HIP* hippocampus, *MCC* midcingulate cortex, *MNI* Montreal Neurological Institute, *PCC* posterior cingulate cortex, *PHIP* parahippocampus, *pINS* posterior insula, *ROI* regions of interest, *SOM* somatic, *VISC* visceral, *vlPFC* ventrolateral prefrontal cortex, *vmPFC* ventromedial prefrontal cortex.
[a]Results also significant against conjunction null).

to $\Delta$CS$^{+}_{SOM}$, Fig. 3d, Table 4). In contrast, other groups revealed no differences between modalities in differential neural activation induced by interoceptive compared to exteroceptive threat predictors at all (Tables 5 and 6).

## Discussion

Adaptive human behaviour in complex environments with multiple threats is guided by evolutionary-driven survival strategies that are preserved across species. In the face of imminent threat, learning from experience is particularly fundamental to the ability to identify and remember predictors of danger to facilitate avoidance or escape. As a translational model at the interface of psychology and the neurosciences, Pavlovian conditioning has proven valuable to elucidating behavioural and neural mechanisms underlying conditioned fear during the expectation of threat[49–51], with widely

appreciated clinical implications for anxiety and stress-related disorders[1]. Herein, we broadened the scope to unravel learning and memory processes underlying negative expectations in the face of multiple threats from different sensory modalities, with a particular focus on the pain. Pain is a ubiquitous and highly salient threat and a crucial part of the organism's survival system that evokes strong adaptive responses, including cognitive and emotional processes orchestrated within the brain. These guide behaviour not only in response to actual pain experience[52], but more importantly also during pain expectation[8,53,54]. Interoceptive, visceral pain appears to be particularly threatening[4,5], engages partly distinct neural representations[37], and may have a specific functional role in shaping brain dynamics[27]. Given the evolutionary significance of aversive signals originating from within our bodies, interoceptive conditioning could evoke greater and more persisting conditioned

**Table 5 Neural activation induced by cues (CS) predicting interoceptive versus exteroceptive threats during reinstatement-test after somatic single threat reinstatement in study 1.**

| Contrast | Region | MNI-coordinates | | | | t-value | P |
|---|---|---|---|---|---|---|---|
| | | H | x | y | z | | |
| Study 1: US$_{SOM}$-subgroup ('single threat reinstatement'; N = 20) | | | | | | | |
| ΔCS$^+_{VISC}$ ∩ ΔCS$^+_{SOM}$ | | | | | | | |
| ROI analyses[1] | pINS | R | 38 | −16 | 22 | 3.16 | 0.004 |
| Whole-brain analyses[1] | SMA | L | −14 | −2 | 48 | 3.94 | <0.001 |
| | Superior frontal gyrus | R | 22 | −12 | 68 | 3.58 | <0.001 |
| | Superior frontal gyrus (dmPFC) | R | 22 | 62 | 6 | 2.58 | <0.001 |
| | Middle frontal gyrus (vlPFC) | L | −32 | 26 | 50 | 2.56 | <0.001 |
| | | R | 44 | 48 | 16 | 2.57 | <0.001 |
| | Inferior frontal gyrus, triangular (vlPFC) | R | 54 | 34 | 6 | 3.20 | <0.001 |
| | Inferior frontal gyrus, opercular (vlPFC) | R | 48 | 10 | 12 | 2.84 | <0.001 |
| | Rectus (vmPFC) | L | −4 | 24 | −22 | 3.19 | <0.001 |
| | | R | 12 | 40 | −22 | 2.57 | <0.001 |
| | Precentral gyrus | L | −16 | −18 | 70 | 2.82 | <0.001 |
| | Superior temporal gyrus | L | −62 | −16 | 10 | 2.18 | <0.001 |
| | | R | 68 | −32 | 16 | 2.86 | <0.001 |
| | Temporal pole: superior temporal gyrus | L | −40 | 20 | −24 | 2.60 | <0.001 |
| | Middle temporal gyrus | R | 68 | −28 | −2 | 3.03 | <0.001 |
| | | R | 36 | 18 | −30 | 3.86 | <0.001 |
| | Rolandic operculum | L | −44 | −16 | 18 | 2.55 | <0.001 |
| | | R | 42 | −16 | 22 | 3.52 | <0.001 |
| | Inferior parietal gyrus | R | −48 | −38 | 42 | 2.21 | <0.001 |
| | Superior occipital lobe | L | −18 | −68 | 28 | 2.55 | <0.001 |
| Whole-brain analyses[2] | Superior frontal gyrus, orbital (vmPFC) | R | 16 | 58 | −10 | 3.00 | <0.001 |
| ΔCS$^+_{VISC}$ > ΔCS$^+_{SOM}$ | – | – | – | – | – | – | – |
| ΔCS$^+_{VISC}$ < ΔCS$^+_{SOM}$ | – | – | – | – | – | – | – |

Shared and differential neural activation induced by predictors of interoceptive (CS$^+_{VISC}$) and exteroceptive threat (CS$^+_{SOM}$) relative to safety-predictive CS$^−$, during reinstatement-test (RST-TEST) after single threat reinstatement with US$_{SOM}$. For shared activation, results of conjunction analyses against global null are presented ([1]{CS$^+_{VISC}$ > CS$^−$} ∩ {CS$^+_{SOM}$ > CS$^−$}; [2]{CS$^+_{VISC}$ < CS$^−$} ∩ {CS$^+_{SOM}$ < CS$^−$}). For differential activation, results of second-level paired t-tests are presented ({CS$^+_{VISC}$ < CS$^−$} > {CS$^+_{SOM}$ < CS$^−$} and vice versa). Peak voxel indicates results of ROI analyses (cluster size $k_E$ ≥ 3; all $P_{FWE}$ < 0.05) and whole-brain analyses (*in italic font;* cluster size $k_E$ ≥ 10; all $P_{uncorrected}$ < 0.001). Exact unilateral P-values are provided. For a visualisation, see Fig. 3 and Supplementary Fig. S7.
*CS* conditioned stimuli, *FWE* family-wise error, *dmPFC* dorsomedial prefrontal cortex, *H* hemisphere, *MNI* Montreal Neurological Institute, *pINS* posterior insula, *ROI* regions of interest, *SMA* supplementary motor area, *SOM* somatic, *US* unconditioned stimuli, *VISC* visceral, *vlPFC* ventrolateral prefrontal cortex, *vmPFC* ventromedial prefrontal cortex.

responses relevant to nocebo mechanisms underlying hypervigilance and hyperalgesia.

In two independent fMRI studies, we implemented specific, yet complementary differential conditioning paradigms to elucidate the acquisition and extinction of conditioned responses to predictors of interoceptive and exteroceptive threats. Experimental visceral pain as clinically-relevant interoceptive US was significantly more unpleasant when compared to the exteroceptive US from two sensory modalities, i.e. exteroceptive somatic pain in study 1 and aversive tone in study 2, despite careful matching to intensity and unpleasantness, respectively, supporting possible differences in habituation processes[4]. In addition to shared neural activation induced by US of different modalities, interoceptive US interestingly evoked greater neural activation within the anterior insula and dorsal anterior cingulate cortex as key regions of the salience network, with well-established roles in the central integration of interoceptive sensory signals with emotional and cognitive facets[55–57]. These findings, which appeared robust even when considering differences in US ratings as nuisance variables, reproduce and complement earlier efforts to elucidate the specificity of interoceptive visceral pain in shaping aversive anticipation and central pain processing, not only in direct comparison to an exteroceptive painful threat[4,26,58], but also to a non-nociceptive, yet a priori equally aversive auditory threat. In line with a notable recent publication detailing a multivariate brain measure, the Neurologic Pain Signature (NPS), for visceral and somatic stimulation across different independent datasets[37], our results from two independent studies support the unique salience of interoceptive pain as a US, above and beyond specific yet highly intertwined perceptual characteristics of intensity and unpleasantness, and underscore the suitability of this experimental model to elucidating the role of threat modality in associative learning and extinction processes in a clinically-relevant context.

To assess whether US modality distinctly shapes the formation of learned negative expectations, we accomplished analyses of differential conditioned responses to modality-specific threat predictors (CS). Results for the acquisition phases of both studies showed enhanced differential behavioural and neural responses to interoceptive threat predictors, suggesting preferential learning for the visceral modality. This was supported at the behavioural level by greater increases in the negative valence of interoceptive versus exteroceptive predictive cues, which were observed despite comparable contingency awareness, and greater visceral cue-induced SCR responses suggested by exploratory analyses of a subset of data. Within the brain, we documented shared differential activation to all threat predictors compared to safety cues in highly overlapping brain regions across studies, in line with a recent meta-analysis documenting a consistent and robust pattern of an 'extended fear network' across diverse fear conditioning paradigms[50], as well as a meta-analysis supporting that pain-related and non-pain-related conditioned fear recruits overlapping but distinguishable neural networks[24]. Importantly, we also consistently demonstrated differences between interoceptive versus exteroceptive threat predictors in both studies. Specifically, conditioned interoceptive threat predictors induced greater differential neural activation in the posterior insula and midcingulate cortex. A recent meta-analysis focusing on pain anticipation supported an interplay of insular and cingulate regions in the representation of the affective qualities of sensory

**Table 6 Neural activation induced by cues (CS) predicting interoceptive versus exteroceptive threats during reinstatement-test after multiple threat reinstatement in study 2.**

| Contrast | Region | MNI-coordinates | | | | t-value | P |
|---|---|---|---|---|---|---|---|
| | | H | x | y | z | | |
| Study 2: RST with US$_{VISC}$ and US$_{SOM}$ ('multiple threat reinstatement'; N = 23) | | | | | | | |
| $\Delta CS^+_{VISC} \cap \Delta CS^+_{AUD}$ | | | | | | | |
| ROI analyses[1] | aINS | R | 34 | 6 | 14 | 2.94 | 0.011 |
| | pINS | L | −32 | −24 | 8 | 3.52 | <0.001 |
| | dACC | L | 0 | 10 | 40 | 2.47 | 0.009 |
| | MCC | L | −4 | −18 | 32 | 2.70 | 0.032 |
| | | R | 14 | −20 | 40 | 3.28 | 0.004 |
| | HIP | L | −30 | −32 | −2 | 3.72 | <0.001 |
| | | R | 28 | −38 | 0 | 3.14 | 0.001 |
| Whole-brain analyses[1] | Inferior frontal gyrus, triangular (dlPFC) | L | −44 | 12 | 24 | 3.03 | <0.001 |
| | ACC | L | 2 | 36 | 28 | 3.21 | <0.001 |
| | PCC | R | 12 | −46 | 20 | 3.05 | <0.001 |
| | Thalamus | L | −20 | −28 | 10 | 5.20 | <0.001 |
| | | R | 14 | −24 | 14 | 4.62 | <0.001 |
| | Middle temporal gyrus | L | −60 | −48 | 4 | 3.05 | <0.001 |
| | Precuneus | L | −2 | −58 | 30 | 3.47 | <0.001 |
| | Paracentral lobe | R | 10 | −34 | 50 | 2.13 | <0.001 |
| | Gyrus fusiformis | L | −30 | −66 | −6 | 2.38 | <0.001 |
| | | R | 38 | −56 | −14 | 2.63 | <0.001 |
| | Vermis | R | 4 | −36 | −16 | 2.33 | <0.001 |
| | Cerebellum | R | 26 | −52 | −22 | 2.95 | <0.001 |
| Whole-brain analyses[2] | Middle temporal gyrus | R | 52 | −58 | 12 | 3.38 | <0.001 |
| | Supramarginal gyrus (S2) | R | 48 | −44 | 22 | 2.41 | <0.001 |
| | Calcarine fissure | R[a] | 28 | −58 | 12 | 2.61 | <0.001 |
| $\Delta CS^+_{VISC} > \Delta CS^+_{AUD}$ | – | – | – | – | – | – | – |
| $\Delta CS^+_{VISC} < \Delta CS^+_{AUD}$ | – | – | – | – | – | – | – |

Shared and differential neural activation induced by predictors of interoceptive (CS$^+_{VISC}$) and exteroceptive threat (CS$^+_{AUD}$) relative to safety-predictive CS$^-$, during reinstatement-test (RST-TEST) after multiple threat reinstatement with US$_{VISC}$ and US$_{AUD}$. For shared activation, results of conjunction analyses against global null are presented ([1]{CS$^+_{VISC}$ > CS$^-$} ∩ {CS$^+_{AUD}$ > CS$^-$}; [2]{CS$^+_{VISC}$ < CS$^-$} ∩ {CS$^+_{AUD}$ < CS$^-$}). For differential activation, results of second-level paired t-tests are presented ({CS$^+_{VISC}$ < CS$^-$} > {CS$^+_{AUD}$ < CS$^-$}; and vice versa). Peak voxel indicates results of ROI analyses (cluster size $k_E \geq 3$; all $P_{FWE} < 0.05$) and whole-brain analyses (in italic font; cluster size $k_E \geq 10$; all $P_{uncorrected} < 0.001$). Exact unilateral P-values are provided. For a visualisation, see Fig. 3 and Supplementary Fig. S7.
aINS anterior insula, CS conditioned stimuli, FWE family-wise error, dACC dorsal anterior cingulate cortex, dlPFC dorsolateral prefrontal cortex, H hemisphere, HIP hippocampus, MCC midcingulate cortex, MNI Montreal Neurological Institute, pINS posterior insula, ROI regions of interest, S2 secondary somatosensory cortex, VISC visceral.
[a]Results also significant against conjunction null.

events, particularly applying to interoceptive signals[59], in line with our own recent findings documenting the relevance of posterior insula in visceral compared to somatic pain expectation[26]. Given the well-established role of the posterior insula in restoring and maintaining homoeostasis in the face of imminent danger[60], its distinct involvement may serve adaptive modulatory functions during the expectation of interoceptive threat. Our findings extend knowledge from other brain imaging studies on the modality-specific aversive expectancy that have compared predictors of somatic pain with aversive pictures[31] or disgusting odours[30], and complement our own data on nocebo effects and underlying mechanisms in visceral pain[8,9,11,21]. These observations suggest a specific relevance of insular together with cingulate regions in the preferential acquisition of the presumably more salient interoceptive CS–US association. In keeping with the notion that expectations dynamically influence perception and learning[61,62], the reciprocal impact of interoceptive threats and their predictors is further substantiated by our exploratory correlational results. These not only indicate that affective qualities of interoceptive versus exteroceptive threats shape conditioned negative expectations, but also suggest a tight link between differential neural responses during the expectation and experience of aversive interoceptive signals. Together, our findings regarding the formation of negative interoceptive expectancies by aversive conditioning support our hypothesis that in the face of multiple danger signals indicating bodily harm, visceral pain evokes preferential interoceptive fear learning. These findings could be viewed as a modern replication of classical

interoceptive conditioning studies carried out by soviet psychologists[35], complemented herein by brain imaging techniques. They are in keeping with preparedness theory[40], and support its applicability to pain-related learning in a broader context of the affective neurosciences, with intriguing putative clinical relevance. The role of fear and hypervigilance is increasingly appreciated in the pathophysiology and treatment of multiple complex and over-lapping clinical conditions, including anxiety and chronic pain[54]. Modality-specific conditioning could therefore contribute to unravelling nocebo mechanisms relevant to vulnerability to chronicity and treatment failure, especially when nocebo effects persist rather than extinguish.

Persisting or resurging fear constitutes a core target of cognitive-behavioural treatment approaches like exposure therapy, which is essentially built on the successful and robust extinction of conditioned responses including learned fear. When the threat is no longer present, extinction of conditioned responses to former threat predictors is adaptive, allowing behavioural flexibility in rapidly changing, complex environments. At the same time, the initially acquired memory trace is preserved and can be dynamically reactivated[63], which can contribute to impaired extinction efficacy and to relapse in clinical contexts[64]. This may be particularly relevant for highly salient and fear-evoking threats that are crucial to avoid, like interoceptive pain, as essentially already suggested by the Soviet pioneers of classical conditioning[35]. To elucidate threat modality-specific extinction processes and their underlying neural mechanisms, we tested whether the visceral

CS–US association is more resistant to extinction and more susceptible to memory reactivation or 'relapse', induced by unexpected US exposure (i.e. reinstatement). In an effort to model different aspects of extinction learning, including the clinical reality of patients with waxing and waning symptoms, we herein implemented different experimental extinction and reinstatement protocols. Interestingly, when omission of the US occurred directly after acquisition on the same day in study 1, behavioural results indicated persisting conditioned fear in response to visceral but not somatic pain predictors, despite comparable contingency awareness. While these results may indicate a greater resistance to extinction specifically for the interoceptive CS–US association, a cautious interpretation is warranted given the small number of extinction trials and an overall overestimation of reported CS–US contingency awareness. In study 2, with an extinction phase accomplished on a subsequent study day and more extinction trials, conditioned behavioural responses were no longer evident to either predictive cue, not even in a supplementary analysis of a smaller number of extinction trials, and hence do not support a modality-specific resistance to extinction. However, given that our earlier conditioning work repeatedly documented rapid and full extinction in 1-day paradigms with visceral threats only[20,65], together the present findings could hence also indicate that full extinction of conditioned emotional responses to multiple threats requires more unreinforced trials, especially for threats of higher salience and immediate extinction learning. In light of increasing knowledge regarding the role of consolidation and reconsolidation in the context of conditioned fear[66,67], our divergent findings in studies 1 and 2 call for more mechanistic studies on the temporal dynamics and boundary conditions of pain-related extinction learning in multi-day and multi-threat paradigms, ideally including objective, physiological measures derived from electrodermal activity or pupillometry recordings. Regarding brain imaging results for the extinction phase, no differences were observed in neural responses to threat predictors of different modalities in either study. Instead, shared neural activation induced by both threat predictors during extinction was evident in both studies, involving key areas of the extinction network, particularly the hippocampus, supporting its general role in extinction learning[68] irrespective of threat modality, or of the number and timing of extinction trials.

Although extinction efficacy is clearly relevant to chronicity and treatment failure in patients[69,70], underlying mechanisms remain incompletely understood even in healthy individuals. Reinstatement constitutes a promising translational tool[38], which has not been applied in human brain imaging studies in the context of multiple threats. While we implemented different reinstatement procedures in the two studies, all involved unexpected US exposure followed by unpaired cue presentations. This allowed us to analyse differential conditioned responses during a reinstatement-test phase for different (former) threat predictors as an indicator of extinction efficacy. Our behavioural findings provide at least partial support for the notion that extinction efficacy may be reduced for the interoceptive CS–US association, i.e. that reinstatement with the visceral US had a greater impact on differential responses to CS. After reinstatement involving unexpected exposure to the visceral US, either as a 'single threat' ($US_{VISC}$-subgroup of study 1) or as a 'multiple threat' (all participants in study 2), we observed greater negative valence of former interoceptive versus exteroceptive threat predictors. On the other hand, reinstatement with the somatic US alone ($US_{SOM}$-subgroup of study 1) did not induce differences between CS modalities. Our hypothesis is most clearly supported by the results of study 2, in which reinstatement induced a selective resurgence of the interoceptive CS–US association, consistent with a return of interoceptive fear. Interpretation of findings in study 1 is complicated by the fact that extinction was immediate and shorter, as explained above, and evidently did not lead to a complete resolution of conditioned responses. Herein, reinstatement can rather be conceptualised as a disruption of the ongoing extinction process, which would make a return (i.e. a de novo increase) of conditioned responses difficult to detect. While we did not plan for this, and results may be hampered by limited statistical power given relatively small sample sizes of reinstatement groups, applicability to real-life scenarios is intriguing: Unexpected and unsignaled threats like painful episodes can obviously occur at any time point during an ongoing extinction process. Understanding how easily such a process can be disrupted would hence inform our understanding of adaptive extinction learning with relevance to factors that may interfere with successful exposure-based treatment. Given these considerations, one interpretation of data from study 1 is that unexpected exposure to interoceptive threat disrupted or in fact halted the ongoing extinction of conditioned responses to interoceptive threat predictors, whereas the extinction process for the same predictors effectively continues after unexpected exposure to exteroceptive threat, as supported by within-modality comparisons. Interestingly, residual effects of cue aversiveness have been shown to predict a reinstatement effect in healthy volunteers[71], consistent with evidence from a clinical setting[72]. Hence, residual fear responses after extinction may serve as a predictor of treatment outcome. In patients with persistent fear and/or pain, achieving a robust and sustained extinction of conditioned responses constitutes an important treatment goal, and maybe particularly challenging for interoceptive memory traces based on our data in healthy individuals. In other words, we may be primed to preferentially learn, store, and remember cues that signal internal harm.

Within the brain, after reinstatement with visceral US alone or in combination with the somatic US, shared differential neural responses to both threat-predictive cues were consistently observed in the hippocampus as a core region of the extinction network. This finding is well in line with earlier studies implementing only one threat modality[73–75], including visceral pain in healthy individuals[76] and in patients with chronic visceral pain[77,78]. In addition to the hippocampus, posterior insula and cingulate regions were differentially activated following reinstatement with visceral threats alone, a neural activation pattern that closely resembled neural responses detected during the acquisition, yet not observed during extinction. These enhanced differential responses within the insula and cingulate cortex during reinstatement-test may reflect the reactivation of the excitatory memory trace, presumably triggering preparatory responses in expectation of the reoccurrence of threat. Notably, single threat visceral reinstatement (study 1), resulted in enhanced differential neural responses for interoceptive compared to the exteroceptive threat predictors, whereas multiple threat reinstatement (study 2) led to shared differential activation of these regions to both interoceptive and exteroceptive threat cues. The latter finding may be explained by generalisation effects induced by the unexpected exposure to multiple threats in close temporal proximity. This may promote a generalisation of threat value from the more salient interoceptive to the less salient exteroceptive threat, ultimately resulting in the reactivation of neural responses to all former threat predictors regardless of their salience. While this is speculative, the inability to adequately differentiate between stimuli of different threat value, particularly when confronted with recent adversity, has previously been discussed as one mechanism contributing to enhanced relapse risk in clinical populations undergoing extinction-based treatment[74]. Together, these observations underline the critical importance of factors promoting either discrimination or generalisation of

conditioned responses in the prediction and prevention of fear relapse[79,80], extending the concept that as a result of conditioning, impaired discrimination of conditioned[81] and unconditioned responses[82] could play a role in the development of chronic pain.

Conditioned interoceptive fear should generally be considered adaptive, and an essential component of evolutionary-driven survival behaviour. However, when contextualised within a broader nocebo framework in which conditioned negative expectations drive maladaptive avoidance, hypervigilance and hyperalgesia[8,83,84], putative clinical implications and future directions are noteworthy. Our findings support that interoceptive threat predictors may more readily evoke conditioned fear, which could drive the transition from acute to chronic pain as well as symptom chronicity, especially in vulnerable individuals. Furthermore, the risk for impaired extinction efficacy and relapse phenomena may be more pronounced in the context of aversive interoceptive signals, especially in combination with stress[83,85], which demonstrably amplifies visceral nocebo effects[86], and may contribute to a negative recall bias about aversive visceral experiences[87]. Given the evidence supporting altered extinction learning in patients with chronic pain, including IBS[77,78], translational research in clinical populations is urgently needed. While our results are limited by the lack of complete SCR data as a biological marker of learning as well as by a more definite exclusion of pain-modality-specific vascular artefacts induced by gasping or other respiratory or movement-related effects that could be more closely inspected whether pulse oximetry or respiration had been measured, they do provide a more refined understanding of conditioned nocebo effects in the context of clinically-relevant interoceptive and exteroceptive threats. Merging our clinically-driven perspective with the rapidly expanding general literature on interoception and predictive processing provides opportunities for the development or refinement of computational models based on the precision of interoceptive versus exteroceptive signals, advancing not only the definition of salience itself but also clarification its mechanistic basis[88–93]. The translation of this knowledge may ultimately help understand and minimise negative expectancy effects in patients with chronic pain[94,95], especially in disorders of gut–brain interactions, adds a brain perspective to the eloquent claim that the gut is 'smart' due to its capability to learn and remember[96], and supports further efforts towards extinction-based treatment approaches for these highly prevalent conditions[10,97].

## Methods

**Participants**. For the purposes of this report, we analysed unpublished data from healthy volunteers who were recruited to serve as controls in two conceptually connected fMRI conditioning studies conducted within a collaborative research unit (SFB 1280 'Extinction Learning', funded by the German Research Foundation). We utilised data from healthy volunteers recruited as part of a patient study (study 1), and included data from the placebo arm of a pharmacological study (study 2; German Clinical Trials Register, registration ID: DRKS00016706). Recruitment and screening of all healthy volunteers in both studies followed highly-standardised and established procedures in our line of visceral pain research[4,86]. An initial structured telephone screening was followed by a personal interview and a medical examination. Interview and examinations were accomplished in a medically-equipped room within a clinical research unit at the University Hospital Essen, Germany. Exclusion criteria common to both studies were <18 or >45 years of age, body mass index (BMI) < 18 or >30, and any known medical condition or regular medication use (except thyroid medication and hormonal contraceptives). The usual exclusion criteria for magnetic resonance imaging (MRI) applied, and structural brain abnormalities were ruled out upon structural MRI. Perianal tissue damage (e.g. haemorrhoids, fissures), which may interfere with rectal balloon distensions were excluded by digital rectal examination. Pregnancy was excluded with a commercially available urinary pregnancy test (Biorepair GmbH, Sinsheim, Germany) on the day of the experiment. Prior participation in any previous or other ongoing studies involving pain-related conditioning was also exclusionary. Standardised questionnaires were used to screen for recent gastrointestinal complaints[98], symptoms of depression or anxiety (Hospital Anxiety and Depression Scale, HADS)[99], as well as to confirm right-handedness[100,101]. As part of a comprehensive psychosocial questionnaire battery, chronic perceived stress was also assessed (Trier Inventory of

Chronic Stress, TICS)[102]. All the participants reported normal hearing and normal or corrected-to-normal vision. The work was conducted in accordance with the Declaration of Helsinki, and studies were approved by the ethics committee of the University Hospital Essen (protocol numbers 10–4493 and 16–7237), and followed the relevant ethical guidelines and regulations. All volunteers provided written informed consent and were paid for their participation.

**Overview of study designs and procedures**. To elucidate the formation and extinction of conditioned responses to threat-predictive cues (conditioned stimuli, CS) in the face of multiple different biologically-salient threats, we implemented two differential delay conditioning studies with visual CS$^+$ predicting interoceptive threat (visceral pain: US$_{VISC}$) or exteroceptive threat (study 1: somatic thermal pain, US$_{SOM}$; study 2: aversive auditory stimulus, US$_{AUD}$, study 2), and unpaired CS$^-$. All experimental procedures were conducted in the MRI-suite of the Institute of Diagnostic and Interventional Radiology and Neuroradiology at the University Hospital Essen, Germany. In both studies, perceptual thresholds for each US modality were initially assessed, and individual US stimulus intensities for implementation during conditioning were identified with a calibration and matching procedure (for details, see below). During conditioning, both studies implemented the same sequence of experimental learning phases, namely acquisition, extinction, and reinstatement-test phases (Fig. 4, details below). In study 1, all phases were accomplished consecutively on a single study day, whereas in study 2, extinction and reinstatement-test phases were implemented 24 h after acquisition. During all learning phases, fMRI was applied to assess shared and differential neural activation induced by US and CS. US- and CS-related behavioural measures were acquired for each phase using digitised visual analogue scales (VAS). Moreover, electrodermal activity was continuously recorded aiming for analysis of skin conductance responses (SCR) as a psychophysiological measure of learning using an MRI-compatible system (Biopac Systems, Inc., Goleta, CA, USA; MP100 in study 1, MP160 in study 2), but technical difficulties resulted in incomplete data. Results of exploratory analyses of skin conductance responses to CS for a subset of participants in studies 1 and 2 are provided as Supplementary Material (Supplementary Fig. S1). Note that participants were informed that the study goals were to investigate neural mechanisms underlying visceral pain-related learning and memory processes. Importantly, no detailed information was provided about experimental phases, or about the contingencies between CS and US.

**Unconditioned stimuli (US)**. For interoceptive US (US$_{VISC}$), applied in both studies, pressure-controlled rectal distensions were carried out with a barostat system (modified ISOBAR 3 device, G & J Electronics, Toronto, ON, Canada). Graded distensions of the rectum with an inflatable balloon constitute a well-established experimental model to assess visceroception and visceral pain, especially in the context of IBS[103]. The distension model allows the controlled and finely-tuned application of distensions inducing mild, intermediate or strong sensations of urgency, discomfort, and pain that closely resemble aversive visceral sensations experienced by patients, which are also commonly but less frequently experienced by healthy persons. Building on our long-standing experimental expertise with different sensory modalities[4,47,48,86], for the exteroceptive US, cutaneous thermal stimuli (US$_{SOM}$) were applied on the left ventral forearm with a thermode (PATHWAY model CHEPS; Medoc Ltd. Advanced Medical Systems, Ramat Yishai, Israel) in study 1. In study 2, an aversive tone (US$_{AUD}$) with a saw-tooth waveform profile and a frequency of 1 kHz created using Audacity 1.3.10-beta (http://www.audacity.sourceforge.net/) was presented by an MRI-compatible sound system (Amplifier mkll+S/N 2016-2-2-03, MR confon GmbH, Magdeburg, Germany) bi-aurally through headphones. Individual perceptual thresholds for US are reported herein only for the purpose of descriptive characterisation of the study samples, as they primarily served as anchors for US calibration and matching.

In a continuation of our previous work on specificity to pain modality[4,26,86], US$_{VISC}$ and US$_{SOM}$ were matched to perceived pain intensity in study 1. In study 2, US$_{VISC}$ were compared to a non-nociceptive, yet equally aversive US$_{AUD}$ by matching to perceived unpleasantness. In both studies, visceral stimuli served as an anchor for calibration and matching of exteroceptive, aiming to identify individual US stimulation intensities within a predefined perceptual range of 60–80 mm (assessed on 0–100 mm VAS, ends labelled not painful and extremely painful in study 1, and not unpleasant and very unpleasant in study 2). To this end, a distension pressure 5 mmHg below the individual rectal pain threshold was initially chosen and rated on VAS until pressure within the predefined range was identified. This was successfully accomplished in both studies (VAS ratings for US$_{VISC}$: 70.1 ± 0.9 mm in study 1; 65.8 ± 3.6 mm in study 2). For matching, visceral stimuli were presented with thermal (study 1) or auditory stimuli (study 2), respectively, and participants were prompted to compare the stimuli on a response device with Likert-type response options indicating more, less, or equally painful (study 1) or unpleasant (study 2) stimuli. If the rating showed a deviation, the intensity of exteroceptive stimuli was successively adjusted until ratings indicated equal perception at least twice consecutively. Of note, stimulus durations for interoceptive and exteroceptive US were adjusted for each individual, aiming at matched durations of ascending and plateau phases of US stimulation (20 s study 1; 14 s study 2). For additional details, see Supplementary Methods.

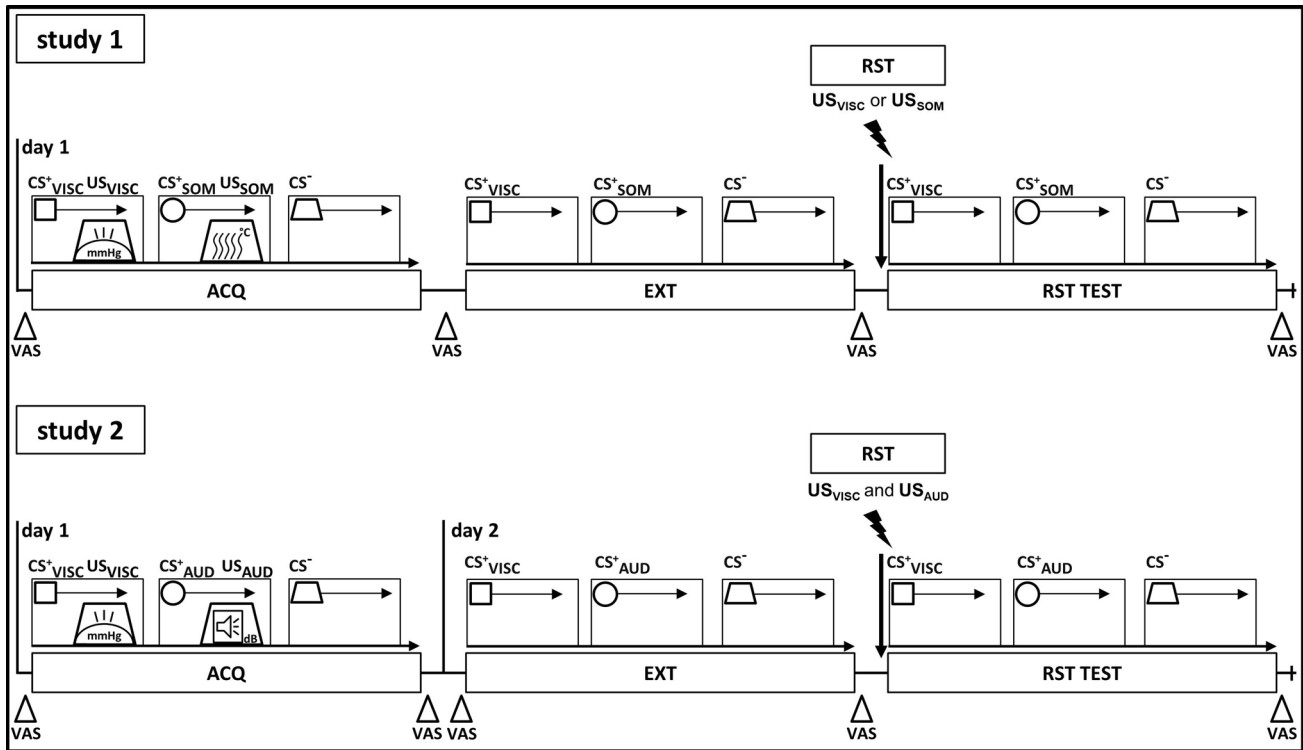

**Fig. 4 Schematic overview of study designs.** All participants in studies 1 and 2 underwent acquisition (ACQ), extinction (EXT), and reinstatement-test (RST-TEST) phases. As for unconditioned stimuli (US), visceral pain (US_VISC) and either equally painful somatic pain (study 1, US_SOM), or equally-unpleasant auditory stimuli (study 2, US_AUD) were implemented during acquisition (ACQ) and reinstatement (RST). As conditioned stimuli (CS), distinct visual geometrical symbols were paired with US (CS+_VISC; CS+_SOM; CS+_AUD) or were presented without US (CS−) during acquisition (differential delay conditioning). All CS were presented without US during EXT and RST-TEST. RST procedures involved unsignalled US from one modality ('single threat reinstatement' in study 1: US_VISC in one subgroup; US_SOM in another subgroup) or from both modalities (multiple threat reinstatement in study 2: US_VISC and US_AUD in all participants). During all phases, functional magnetic resonance imaging (fMRI) was accomplished to assess shared and differential CS- and US-induced neural activation in regions of interest. Before and after each phase, behavioural measures were acquired with visual analogue scales (VAS).

Note that after matching, in study 1 a short adaptation phase was accomplished in the MR scanner to accommodate for possible habituation effects previously observed for thermal pain stimuli[4]. This involved a short series of unsignaled US presentations (i.e. five visceral and five heat pain in pseudorandomized order), followed by another matching procedure when necessary. Supplementary control analysis of movement data (linear and degree movement) was accomplished for the unsignaled US delivered in the habituation phase conducted prior to the acquisition, in order to explore possible pain-modality-specific movements (e.g. due to gasping) potentially confounding differential neural activation in subsequent experimental phases (Supplementary Fig. S3).

Based on these careful matching procedures, the following stimulus intensities for implementation during acquisition and reinstatement were identified: For US_VISC distension pressures, $34.7 \pm 1.6$ mmHg in study 1, $35.9 \pm 1.8$ mmHg in study 2; for US_SOM thermode temperature, $45.1 \pm 0.3$ °C; for US_AUD loudness, $94.74 \pm 1.18$ dB SPL (range: 89–108 dB SPL), all in line with our earlier results involving the application of the same pain[4,86] or auditory stimuli[47,48].

**Experimental phases.** During acquisition, both studies involved three distinct conditioned stimuli (CS), which were contingently paired with the interoceptive US (CS+_VISC) and one exteroceptive US (CS+_SOM or CS+_AUD, respectively), or remained unpaired (CS−). Visual geometric symbols served as CS, and allocation of a specific CS symbol to a specific US (US_VISC, US_SOM, or US_AUD) or designation as CS− was counterbalanced across participants. Within each study, participants were pseudo-randomly assigned to a different order sequence of CS–US pairings to avoid potential sequence effects. The programming of pairings aimed for an essentially pseudorandomized order, but avoided more than two successive pairings of one modality, and ensured that sequences alternatingly started with an interoceptive or exteroceptive CS–US pairing. All acquisition sequences were also balanced for the number of CS+ and CS− presentations. The number of CS presentations and reinforcement schedules were similar in the two studies (study 1: 10 presentations per CS, 8 CS–US pairings, 80% reinforcement; study 2: 12 presentations per CS, 10 CS–US pairings, 83% reinforcement). All CS+ were presented 6–12 s before US, with CS and US co-terminating. Inter-stimulus intervals consisted of a black screen with a white frame (durations: 5–8 s).

During extinction and reinstatement-test phases, CS was presented without any US. Given earlier evidence of rapid extinction for CS–US_VISC associations[20,65] and to ensure tolerability of total scanning time for participants in study 1 (all phases accomplished consecutively), the number of extinction and reinstatement-test trials, respectively, was lower (5 presentations per CS) than in study 2 (12 presentations per CS). Subsequent to the extinction phase, reinstatement procedures involving the unsignaled and unexpected re-exposure to the US were accomplished, followed by a reinstatement-test phase, consisting of the same number of CS presentations as during extinction in pseudorandomized order. Given a lack of human reinstatement studies involving the multiple US, and unresolved methodological challenges in the field[38], we implemented different reinstatement procedures in an effort to provide procedure-specific, yet complementary data. In study 1, participants were pseudo-randomly assigned to subgroups undergoing a reinstatement procedure with either interoceptive US alone (4 US_VISC, $N = 22$) or the exteroceptive US alone (4 US_SOM, $N = 20$). By doing so, we aimed to test for reinstatement effects after unexpected re-exposure to threat from one modality (single threat reinstatement) within each reinstatement subgroup. In study 2, all participants ($N = 23$) underwent a reinstatement procedure with both interoceptive and exteroceptive US (3 US_VISC, 3 US_AUD), aiming to test for reinstatement effects after unexpected re-exposure to threats from multiple modalities (multiple threat reinstatement). Note that the number of unexpected US presentations was chosen based on work in the field[38] and our own earlier studies[20,77,104]. All US intensities and durations implemented as part of reinstatement procedures were identical to those applied during acquisition.

**Behavioural measures.** For the purposes of concise and parallelised US- and CS-related behavioural and neural analyses across different sensory modalities in two independent studies, we focussed our behavioural data analysis on unpleasantness ratings as a clinically-relevant indicator of emotional valence. Emotional valence is relevant to all types of threat, shapes the perception of aversive stimuli, including pain[105], and drives threat-related behaviours like approach and avoidance[106]. It is highly relevant to the specificity of visceral pain[26,42], and sensitive to modulation by placebo/nocebo mechanisms[8,10,107,108]. Prior pain-related conditioning studies from our own group (reviewed in refs. [8,109]) and in the broader fear conditioning

literature support the notion that conditioned changes in cue valence constitute a sensitive and relevant behavioural measure capturing the formation, as well as the extinction and return of fear responses in healthy adults[110] and clinical populations[106].

All ratings were accomplished on digitised VAS in the scanner using an MRI-compatible hand-held fibre optic response system (LUMItouchTM, Photon Control Inc., Burnaby, BC, Canada) before and after learning phases (for specific assessment time points, see Fig. 4; note that in study 2, an additional VAS rating was accomplished mid-extinction (after 6 trials), which we report on in the Supplementary Table S7). In study 1, VAS anchors were labelled 'very pleasant' (−100 mm) and 'very unpleasant' (+100 mm), with the word 'neutral' (0 mm) marked in the middle of the digitised VAS, as accomplished in our previous conditioning work involving painful US[20,21,65,77,86,111]. In study 2, VAS anchors were labelled 'not at all unpleasant' (0 mm) and 'very unpleasant' (100 mm), consistent with our prior work across sensory modalities[47,48]. For the purposes of this report and in light of differing scales, we exclusively analysed differential CS valence, computed as individual delta (Δ) scores for each CS$^+$ relative to the CS$^-$ for each learning phase and within each study group. This allows phase-specific comparisons of $\Delta CS^+_{VIS}$ vs. $\Delta CS^+_{SOM}$ in study 1 and $\Delta CS^+_{VIS}$ vs. $\Delta CS^+_{AUD}$ in study 2, in keeping with contrasts computed for brain imaging analyses (see below). Note that we additionally acquired perceived intensity of $US_{VISC}$ and $US_{SOM}$ in study 1; for findings dedicated to elucidating the contributions of intensity versus unpleasantness in the context of visceral pain specificity, see our earlier work[4,26] and Supplementary analyses herein using these ratings as a covariate of no interest for fMRI data analyses (Supplementary Tables S1–2, 9-10-S9).

To elucidate cognitive awareness of the specific CS–US pairings for each phase, we report contingency awareness as a secondary behavioural measure. To this end, at the conclusion of each experimental phase, for each CS a VAS with ends labelled 'never' (0 mm) and 'always' (100 mm) assessed the perceived probability (0–100%) of a US following a specific CS, as previously described[20,77,104]. Note that we herein report contingency awareness with a focus on differences between modalities. Statistical analyses of the accuracy of CS–US associations require more complex computations (for an approach, see ref. [112]), which is beyond the scope herein.

**Statistical analyses and reproducibility of behavioural data**. Statistical analyses of behavioural data were accomplished separately for each study using IBM SPSS Statistics for Windows, version 20 (IBM Corp., Armonk, N.Y., USA). For ΔCS valence, 2 × 2 repeated-measures analyses of variance (rmANOVA) with the factors time (pre, post) and modality (interoceptive, exteroceptive) were computed for each experimental phase, applying the Greenhouse-Geisser correction when the assumption of sphericity was violated. Given our hypotheses and to ensure readability, we provide statistical details on time × modality interaction effects in the main manuscript; full rmANOVA results including all main and interaction effects are given in Supplementary Tables (Supplementary Tables S4 and S5). Two-tailed paired $t$-tests were computed as planned comparisons for two purposes: (1) To test for hypothesis-driven differences between modalities in ΔCS valence, US valence, and contingency awareness at specific time points within studies and experimental groups (all results reported in the main manuscript; further details in Supplementary Tables S6 and S7); and (2) to explore differences in ΔCS valence within modalities across time points (PRE-POST; full results reported in Supplementary Tables S6 and S7). Only Bonferroni-corrected P-values are reported within the main manuscript; full uncorrected results of all paired $t$-tests are provided in Supplementary Tables S6 and S7. For rmANOVA, effect sizes are reported as partial eta squared ($\eta_P^2$); for $t$-tests, effect sizes are provided based on Cohen's d for correlated designs[113]. Correlational analyses were accomplished using Pearson's r. Results are reported as mean ± standard error of the mean (SEM).

All data are expected to be reproducible given the same settings and procedures as described herein.

**Brain imaging data acquisition and analyses**. All MR images were acquired using a whole-body 3 Tesla scanner (Skyra, Siemens Healthcare, Erlangen, Germany) equipped with a 32-channel head coil. For functional imaging, single-shot echo-planar imaging (EPI) sequences with similar settings were used (identical for both studies: TE 28.0 ms, flip angle 90°, GRAPPA r = 2 with 38 transversal slices angulated in the direction of the corpus callosum, slice thickness of 3 mm, slice gap 0.6 mm, voxel size 2.3 × 2.3 × 3.0 mm; study 1: TR 2300 ms, FOV 220 × 220 mm$^2$, matrix 94 × 94 mm$^2$; study 2: TR 2400 ms, FOV 240 × 240 mm$^2$, matrix 104 × 104 mm$^2$). Structural images were acquired prior to functional imaging using the same T1-weighted 3D-magnetisation prepared rapid gradient echo (MPRAGE) sequence in both studies [repetition time (TR) 1900 ms, echo time (TE) 2.13 ms, flip angle 9°, field of view (FOV) 239 × 239 mm$^2$, 192 slices, slice thickness 0.9 mm, voxel size 0.9 × 0.9 × 0.9 mm$^3$, matrix 256 × 256 mm$^2$, Generalised Partially Parallel Acquisitions (GRAPPA) r = 2].

Functional images were analysed with SPM software (SPM12, Wellcome Trust Centre for Neuroimaging, UCL, London, UK) implemented in Matlab (R2016b, Mathworks Inc., Sherborn, MA, USA). A standard realignment procedure as implemented in SPM12 was performed for the estimation of six parameters for translation (linear: x, y, z (mm)) and for rotation (degree: pitch, roll, yaw (°)) to describe the rigid body transformation between each image and a reference image.

Subsequently, functional images were co-registered to individual T1-weighted structural images used as reference images, with the origin set to the anterior commissure. Functional images were normalised to Montreal Neurological Institute (MNI) space using a standardised International Consortium for Brain Mapping (ICBM) template for European brains as implemented in SPM12, and smoothed using an isotropic Gaussian kernel of 8 mm. To correct for low-frequency drifts, a temporal high-pass filter with a cut-off set at 128 s was implemented. Serial autocorrelations were taken into consideration by means of an autoregressive model first-order correction.

First-level analyses were performed using a general linear model applied to the EPI images. The time series of each voxel was fitted with a corresponding task regressor that modelled a box car convolved with a canonical hemodynamic response function (HRF). As regressors, CS type (CS$^+_{VISC}$; CS$^+_{SOM}$/CS$^+_{AUD}$; CS$^-$) and US modality (US$_{VISC}$; US$_{SOM}$/US$_{AUD}$, only in analyses of acquisition phases) were included. For analyses of CS-induced activations, durations were used exactly as implemented in the experiments (jittered between 6 and 12 s before US presentation), for analyses of US-induced activations, ascending and plateau phases of US stimulation were included in analyses (20 s in study 1, 14 s in study 2). Six realignment parameters for translation and rotation were additionally implemented as multiple regressors for motion correction. After model estimation, the following first-level contrasts and respective reverse contrasts were computed for analyses of differential CS-related and US-related neural responses separately for each study group: CS$^+_{VISC}$ > CS$^-$, CS$^+_{SOM}$ > CS$^-$, US$_{VISC}$ > US$_{SOM}$ for study 1; CS$^+_{VISC}$ > CS$^-$, CS$^+_{AUD}$ > CS$^-$, US$_{VISC}$ > US$_{AUD}$ for study 2. CS contrasts were computed for each phase, US contrasts only for the acquisition phase.

On the second level, for analyses of US-induced neural activation, one-sample t-tests based on these differential first-level contrasts and paired t-tests were calculated. For analyses of CS-induced differential neural activation, paired t-tests were computed for each experimental phase to compare $\Delta CS^+_{VISC}$ versus $\Delta CS^+_{SOM}$ in study 1 ({CS$^+_{VISC}$ < CS$^-$} > {CS$^+_{SOM}$ < CS$^-$}; {CS$^+_{VISC}$ > CS$^-$} > {CS$^+_{SOM}$ > CS$^-$}) and $\Delta CS^+_{VISC}$ versus $\Delta CS^+_{AUD}$ in study 2 ({CS$^+_{VISC}$ < CS$^-$} > {CS$^+_{AUD}$ < CS$^-$}; {CS$^+_{VISC}$ > CS$^-$} > {CS$^+_{AUD}$ > CS$^-$}). Additional exploratory analyses included selected covariates of no interest, as indicated in the results. Further, extending our earlier findings revealing not only distinct but also shared neural activations for US[4] as well as CS[26] across modalities, conjunction analyses using first-level contrasts were carried out to identify joint activations (i.e. CS$^+_{VISC}$ > CS$^-$ ∩ CS$^+_{SOM}$ > CS$^-$, US$_{VISC}$ ∩ US$_{SOM}$ for study 1; CS$^+_{VISC}$ > CS$^-$ ∩ CS$^+_{AUD}$ > CS$^-$, US$_{VISC}$ ∩ US$_{AUD}$ for study 2). Conjunction analyses were computed (a) using the minimum statistic to the conjunction null to test for shared activation within all tested subjects, and (b) using the minimum statistics to the global null to test for shared activation within some subjects[114,115]. For correlational analyses exploring associations between differential CS and differential US activation in specific ROI (provided in Supplementary analyses), parameter estimates were extracted for peak-voxels in significant regions of interest (ROIs) as identified by one-sample t-tests.

All analyses focused on a priori defined ROIs of the salience and extinction networks[26,51,55–57,66,68], including the insula (anterior, aINS; posterior, pINS), subregions of the cingulate cortex (midcingulate cortex, MCC; dorsal anterior cortex, dACC), amygdala, hippocampus, and ventromedial prefrontal cortex (vmPFC). All ROI analyses were carried out using unilateral anatomical templates constructed from the WFU Pick Atlas (Version 2.5.2), as implemented in SPM12. Segmentation of the insula (aINS, pINS) and cingulate cortex (dACC, MCC) was accomplished with masks based on the previous literature[116] within the borders of the Wake Forest University (WFU) Pick Atlas. For all reported ROI analyses, family-wise-error (FWE) correction for multiple testing was used with statistical significance set at $P_{FWE} < 0.05$, and coordinates refer to the MNI space. Supplementary whole-brain analyses (uncorrected $P < 0.001$) were additionally carried out (Tables 1, 3–6; Supplementary Tables S1–3, S8–12; Supplementary Figs. S4–S7 for visualisation). Note that the results presented within the main manuscript all focus on ROI analyses unless explicitly specified otherwise.

**Reporting summary**. Further information on research design is available in the Nature Research Reporting Summary linked to this article.

## Data availability
All fMRI data analysed for the current study are available in the neurovault repository (https://neurovault.org/collections/GPPGVZAT/). Behavioural and SCR data are provided in the main manuscript or its Supplementary Information; additional data and information upon request.

## Code availability
No custom code or mathematical algorithms were used in the study. All software used for statistical analyses has been declared in the manuscript.

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

## Acknowledgements
We would like to thank Alexandra Kornowski and Dr Marcel Gratz for their excellent technical support. We also thank Sopiko Knuf-Rtveliashvili, Nelly Hazzan, Dr. Alexandra Labanski, Lea Schliephake and Katharina Krawczyk (née Fleischer) for support in data acquisition. The study was funded by the Deutsche Forschungsgemeinschaft (DFG, German Research Foundation; project number 316803389; SFB 1280, subprojects A10 and A12). The funding organisation was not involved in study design; in the collection, analysis, and interpretation of data; in the writing of the report; or in the decision to submit the article for publication.

## Author contributions
L.R.K., R.J.P. and L.P. performed the research; S.E., H.E., A.I. and L.R.K. designed the research study; L.R.K., R.J.P., L.P., K.F. and N.T. analysed the data; L.R.K., A.I. and S.E. wrote the first draft of the paper; S.E. and H.E. acquired funding; all authors contributed to the interpretation of the data, revised the manuscript for critical content, and approved the final version of the manuscript.

## Funding

## Competing interests
The authors declare no competing interests.
