## [Peer Review File · Communications Biology]

Reviewers' comments:

Reviewer #1 (Remarks to the Author):

The authors propose that classical conditioning of visceral pain has a special status as compared to external signals when used as unconditioned stimuli, producing stronger and longer lasting conditioned responses. To prove this claim, the authors analyze two unpublished fMRI pavlovian learning tasks in which participants learn to associate and then forget arbitrary visual cues with rectal distension or exteroceptive unpleasant signals (thermal pain in study 1, auditory cues in study 2). While several differences exist in the extinction phase between the two datasets (extinction phase is separated into two days in study two and visceral and exteroceptive unconditioned stimuli are reinstated simultaneously), the two datasets are well matched regarding task structure in the acquisition and extinction phase, and stimuli timing and visceral stimulation protocol are virtually identical, making it reasonable to compare between the two studies. Importantly, the level of pain/unpleasantness of each modality was carefully matched in the two studies, such that first, the threshold of visceral pain was obtained and exteroceptive pain or distress was matched to have a perceived, self reported similar intensity, allowing to rule out differences in pain/unpleasantness between the modalities as source of differences behavior and brain activation. fMRI data was analyzed with a series of first and second levels analysis using a priori selected regions of interest consisting of insular cortex, mid and anterior cingulate cortex, hippocampus and amygdala. The effectiveness of the acquisition, extinction and reinstatement is assessed using subjective reports and visual analogue scales, with no objective markers of the effectiveness of the different conditioning stages, which in my opinion constitutes an important limitation to this study.

Their claim of visceral signals having a special status with respect to exteroceptive signals seems to be backed up by their results. Behaviorally, they find a significant difference in reported valence change before and after conditioning for the conditioned stimuli for interoceptive than exteroceptive threats, present in the two studies i.e. arbitrary visual cues predicting painful rectal distension are reported as more negative than those predicting external distress. Furthermore, the self reported association between the conditioned and unconditioned stimuli was more difficult to extinguish in study 1, although not for study 2, the reasons of this discrepancy being ascribed to the fact that in study two, extinction phase was longer and occurred 24 hours later, which seems a reasonable possibility. Neurally, authors report increased brain activation for visceral vs external unconditioned stimuli in anterior insular and dorsal anterior cingulate cortex in both studies, as well as increased activation for interoceptive vs exteroceptive conditioned stimuli in mid cingulate and posterior cingulate cortex.

Overall, the study was carefully designed and executed and the claim goes in line with the obtained results. Regarding the novelty of the results, I have the impression that these results are well expected, based on previous results from the same team (REF 4), or more classical references (REF 5), and as such, are more confirmatory than mechanistic and provide no fundamental insight on the nature of the difference between visceral and somatic stimuli, the computational processes involved, or the recruitment of different brain regions. Furthermore I was expecting a more in depth discussion of the special status of visceral signals, which at several points seems redundant or tautological → e.g. lines 502-503 'Due to its interoceptive nature, visceral pain appears to be particularly threatening' lines 555-556 'the unique salience of visceral pain contributes to preferential interoceptive learning'. What is the uniqueness of visceral signals that makes them more salient? Is it the salience of visceral signals that makes them special or rather the different

anatomical pathways with respect to somatic signals? See Eickhoff 2008 for a classical fmri study on visceral vs somatic fmri activations and Azzalini 2019 for a list of references and a discussion regarding the special status of visceral signals.

My major concern with the paper is regarding the brain figures, which are not able to convey the location of the activation accurately and could potentially greatly confuse the reader (as in my case), raising concerns about the accuracy of the anatomical labeling. Too name a few of this confusing labelings;

-Figure 2A- dACC activation seems to occur in the ventricle. Left AI seems to occur in the brainstem

-2B Left insula activation occurs in the putamen and not in the insula

-Figure 4 Is particularly confusing, hippocampus in panel D and F seems to be in completely different location.

dAcc in panel G in the thalamus. Blob in the left part of panel G is unlabelled.

May I suggest a more standard coronal/sagittal/axial selection of slices, indicating the MNI coordinates and the axes? Or at least the same view for the different contrast?

A second point is the used predefined ROIs. While the selection of ROIs seems justified, several relevant regions are left outside the analysis. Why not including Basal ganglia or secondary somatosensory cortices, which play an important role in either reinforcement learning or visceral pain, as from previous research from the group? Gramsch 2014. Additionally, might I suggest reporting whole brain uncorrected results in the supplementary results, and uploading the unthresholded t maps to neurovault.org, as well as the behavioral data to open science framework? This would increase the reproducibility and trust of the results

Finally, if exteroceptive and interoceptive stimuli were matched in unpleasantness, how come there are significantly greater reported post-acquisition US visceral unpleasantness ratings compared US somatic or auditory? Lines 386-389 are particularly confusing. Furthermore, while authors report observing no differences when using the self-reported unpleasantness data in their model, they do not provide the data while in fact the final, reported models of the paper should definitely include this important covariate.

References

Eickhoff, S. B., Lotze, M., Wietek, B., Amunts, K., Enck, P., & Zilles, K. (2006). Segregation of visceral and somatosensory afferents: an fMRI and cytoarchitectonic mapping study. *Neuroimage*, 31(3), 1004-1014.

Azzalini, D., Rebollo, I., & Tallon-Baudry, C. (2019). Visceral signals shape brain dynamics and cognition. *Trends in cognitive sciences*.

Gramsch, C., Kattoor, J., Icenhour, A., Forsting, M., Schedlowski, M., Gizewski, E. R., & Elsenbruch, S. (2014). Learning pain-related fear: neural mechanisms mediating rapid differential conditioning, extinction and reinstatement processes in human visceral pain. *Neurobiology of learning and memory*, 116, 36-45.

Reviewer #2 (Remarks to the Author):

Review Overview:

Apologies for my late review, as this article came to my desk during the corona crisis.

This is a beautifully conceived and executed paper on an extremely timely topic. In brief, the authors investigated the behavioural and neural effects of interoceptive versus exteroceptive condition stimuli and responses. Using a well-established protocol developed by this lab, they compare the influence of interoceptive (rectal distension) vs exteroceptive (thermal) pain stimuli on subjective valence ratings and fMRI responses during acquisition, extinction, and recall. The study employs a well-controlled and elegant experimental design with numerous positive controls and validation checks. Overall, the results are robust, meaningful, and coherent with what can be expected given previous work in this area. Additionally, I think these results speak to a larger debate in the predictive processing community regarding the role of interoceptive signalling in hierarchical precision weighted inference. I think this paper will have a large impact on the field and generate numerous follow-on behavioral, neural, and clinical studies. My comments consider primarily improving the transparency and reproducibility of these excellent findings, clarifying a few theoretical points, and opening up the discussion to those outside the pain conditioning literature.

Major comments:

1. This is an excellent study. The primary results show that interoceptive conditioned stimuli evoke a greater level of negative valence than matched exteroceptive control stimuli, and that this effect is particularly pronounced during reactivation in the extinction phase. Evolutionarily speaking, this finding makes a lot of sense: errors in the interoceptive modality are almost always life threatening and warrant extreme, 'one-shot' learning. As a simple example, consider how a single case of food poisoning is generally enough to evoke fear and avoidance from that food, and even slightly similar foods, for a lifetime. I think these results speak beautifully to some of these issues, so I just want to congratulate the authors.

2. Along these lines, have the authors looked at the history of interoceptive conditioning literature carried out by the soviet psychologists? I see their experiment very much as a modern replication of these classical results, which is a great thing. In particular, as reviewed by Rezlan (1972, ref#9), the soviet psychologists claimed to find that interoceptive traces were more rapid to acquire, more strongly resisted extinction, and could be reactivated more easily than exteroceptive traces. I think it would be lovely if the authors could contextualize their results in light of these classical findings and note that it detracts absolutely nothing from the novelty here, which is substantive.

3. I really do respect that the authors are very circumspect in their claims and theorizing. Almost all of the results are discussed using relatively theory neutral terms like "salience", couched in the basic language of classical conditioning. In this case, I want to ask the authors to speculate a bit more about these results, in light of the rapidly emerging interoceptive predictive processing literature. A major debate there concerns the "precision" or confidence assigned to visceral signals (both ascending errors and descending predictions). While many have assumed visceral signals are imprecise (ref #), the classical results mentioned in the previous points suggest that deviations on the internal channel seem to be assigned a much higher weight than those on the exteroceptive channel, perhaps as an evolved survival mechanism. This actually speaks to the psychological definition of salience itself, as a weighting of information by whatever yields the best evidence for

ones longterm survival. While I don't expect the authors to go into this level of detail, it would greatly expand the impact of the article if they could connect to this more general literature on interoception and predictive processing. Obviously, without a distinct computational model at hand the authors can at best speculate about the mechanisms at play in their results, but I do believe at least a little speculation is warranted here. See refs 3-8, which could be worked into the discussion at the author's discretion.

4. The authors analyze only subjective valence ratings, which are certainly interesting and at least partially independent from their control of overall pain intensity. However, given the behavioral paradigm I was left wondering why no reaction time data are analyzed. If the interoceptive association is learned more readily, could this not be seen in the RTs themselves? I welcome the subjective data analysis, but it could be strongly complemented by a more behavioral measure of learning, if one is available.

5. I feel like the one weakness of the article is in the neuroimaging results. I am generally not a fan of ROI analyses unless the ROIs have been specifically pre-registered prior to any data collection or analysis, as it is simply too easy to inflate results using this approach. I am not saying the authors did this at all, and I do think their ROIs are sensible given the domain, but it is something I consistently flag as I think we need to be moving away from non-preregistered ROI analyses, generally speaking. With that said, the whole brain results as plotted look a bit odd; many of the 'insula' or 'cingulate' activations look focused in the white matter or CSF. This could potentially imply an issue with nuisance covariance. From my count the authors utilize more than 5 ROIs and 5 contrasts, so the inflation of the family wise false positive error rate could be greatly inflated, even with the FEW peak level correction applied. I think the results should ideally be supplemented with a whole brain analysis.

6. Following on from the above, my primary worry here would be dealing with white noise in the fMRI signal arising from cardiorespiratory fluctuation. Do the authors have any pulse oximetry or respiration belt data? I could easily imagine that the more salient interoception condition might cause "gasping" or increased respiration and heart rate. Ideally, I would like to see a RETROICOR or similar control analysis ensuring that the activation results are not vascular artefacts, which are particularly amplified in the insula and anterior-mid cingulate. At the very least, some analyses of physiological summary statistics would partially assuage these concerns. Additionally, the authors use a sub 2s TR, but applied the default AR(1) technique for whitening. For faster TRs in this range, my understanding is that the SPM authors now recommend the FAST technique, which can better handle aliasing effects that are pronounced at faster acquisition speeds. I think a supplementary control analysis is definitely called for.

7. Openness. This is an excellent article, using a robust experimental design and I applaud the authors for accurately reporting effect sizes alongside statistical tests. However, as a signatory of the COBIDAS guidelines for reproducibility in neuroimaging, I must ask them to go further. Specifically:

7a: Nowhere is there any mention of data and/or code availability. The authors should upload their anonymized behavioral and neuroimaging data to a repository of their choice, and also make available all analysis and paradigm scripts. This is essential for the reproducibility of their important findings. If they are unable to do so, e.g. due to ethical constraints or similar, then it should be explicitly stated within the article why they have chosen not to share these materials, and what steps future authors should take to obtain them. I understand that in many cases it may not be

possible to share raw or subject level biomedical imaging data: in this case, uploading the group level summary t-contrast maps to Neurovault is a sufficient minimum.

7b: Robustness in data visualization. The authors use bar plots in several places to illustrate their behavioral and neuroimaging results, but these can be highly misleading, and obscure the underlying nature of the data. In the interest of transparency and rigor, I encourage the authors to use a more robust plotting approach such as Rainclouds or Beeswarms. See refs # and # for more information and examples.

Minor comments:

1. Page 6, line 159 – here and elsewhere in the manuscript the authors use the phrase “fMRI was accomplished”, but this is incorrect grammar, and an unusual wording. A more correct formulation would be “fMRI was applied to assess..”.

2. Page 15, line 464-465 ‘ the interaction approached significance’. Frequentist tests do not approach, flirt with, or comingle with significance. There are either significant or not, based on the a priori alpha criterion. I don’t believe the authors rest any interpretive weight on these results (if they do so, then they should not), but this phrasing should be revised and generally avoided. If you want a continuous measure of evidence, use a Bayesian or likelihood-based approach.

Finally, I would like to thank the authors for their patience, and their hard work on this excellent manuscript. Following suitable revision I look eagerly forward to its publication.

Sincerely,
Micah Allen

References

1. Allen, M., Poggiali, D., Whitaker, K., Marshall, T. R., & Kievit, R. A. (2019). Raincloud plots: A multi-platform tool for robust data visualization. *Wellcome Open Research*, 4, 63. <https://doi.org/10.12688/wellcomeopenres.15191.1>
2. Eklund, A. (2012). Beeswarm: The bee swarm plot, an alternative to stripchart. *R Package Version 0.1*, 5.
3. Allen, M., Legrand, N., Correa, C. M. C., & Fardo, F. (2020). Thinking through prior bodies: Autonomic uncertainty and interoceptive self-inference. *Behavioral and Brain Sciences*, 43. <https://doi.org/10.1017/S0140525X19002899>
4. Allen, M., Levy, A., Parr, T., & Friston, K. J. (2019). In the Body’s Eye: The Computational Anatomy of Interoceptive Inference. *BioRxiv*, 603928. <https://doi.org/10.1101/603928>
5. Allen, M., & Tsakiris, M. (2018). The body as first prior: Interoceptive predictive processing and the primacy. In *The Interoceptive Mind: From Homeostasis to Awareness* (Vol. 27).
6. Seth, A. K. (2013). Interoceptive inference, emotion, and the embodied self. *Trends in Cognitive Sciences*, 17(11), 565–573. <https://doi.org/10.1016/j.tics.2013.09.007>
7. Onat, S., & Büchel, C. (2015). The neuronal basis of fear generalization in humans. *Nature Neuroscience*, 18(12), 1811–1818. <https://doi.org/10.1038/nn.4166>
8. Grahl, A., Onat, S., & Büchel, C. (2018). The periaqueductal gray and Bayesian integration in placebo analgesia. *eLife*, 7, e32930. <https://doi.org/10.7554/eLife.32930>
9. Razran, G. (1961). The observable and the inferable conscious in current Soviet psychophysiology: Interoceptive conditioning, semantic conditioning, and the orienting reflex. *Psychological Review*,

Reviewer #3 (Remarks to the Author):

In this study, the authors compared acquisition, extinction, and reinstatement of threat in two fMRI studies using visceral vs. non-visceral USs. They found greater responding to visceral USs, and CSs that predicted visceral USs in acquisition and reinstatement in regions of the fear/salience network. This is an interesting study with an adequate sample size and a built-in replication. For the most part, they used robust data collection and analysis techniques (Exceptions discussed below). Given the uniqueness of the study procedures and the rigor of the methodology, I believe that with substantial revisions this manuscript would be of interest to the readers of *Communications Biology*.

Major issues

My primary concern with this manuscript is that the USs were perceived differently after acquisition, which makes it difficult to determine whether their results were due to US intensity or US modality (See 408-410 and 419-421). The authors mention that they have included US rating as a covariate in some of the analyses, but that the data are not included. I would recommend including these data to allow the readers to determine themselves whether the US intensity impacted the findings.

Given that the US intensity was matched at the start of the experiment, a likely candidate for the differences in post conditioning US ratings may be habituation. The authors should discuss the available literature addressing differences in habituation among visceral vs. non-visceral stimuli. The authors included a set of correlations between CS-related activation differences across modalities and US-related differences across modalities (See 429-434). This seems like double-dipping, especially considering the above concerns about post conditioning differences in US intensity ratings.

Throughout the reinstatement section, the authors report the presence/absence of effects in one group vs. another the group x variable interaction (e.g. 460-464). This is not a statistically valid approach. The authors should conduct an omnibus analysis with group as a factor to truly test whether the presence of an effect in one group over another reflects a true significant interaction.

Minor issues

The cutaway figures showing differential activation are distracting and difficult to interpret. With the clusters of activation floating in space, it is difficult to precisely localize their anatomy. If the authors choose to include this figure, they should also include a montage of axial/sagittal/coronal views of highlighted results so that the anatomy can be precisely inspected.

On lines 228-231, the authors mention that the total duration of the US was matched across modalities. This statement is confusing. Were the individual durations variable? How long was each US presented?

For study 2, would make sense to include a supplemental analysis with same number of extinction trials as in study 1 for comparison (See lines 445-449).

If it is possible to include a portion of the SCR data as a supplement, this would be a welcome addition.

COMMSBIO-20-1217A (Revision)

Title: From gut feelings to memories of interoceptive pain:

Associative learning and extinction of conditioned threat predictors across sensory modalities

Authors: L.R. Koenen, R.J. Pawlik, A. Icenhour, L. Petrakova, K. Forkmann, N. Theysohn, H. Engler, S. Elsenbruch

Point-by-point responses to comments:

Comments from Reviewer #1:

Original comment:

The authors propose that classical conditioning of visceral pain has a special status as compared to external signals when used as unconditioned stimuli, producing stronger and longer lasting conditioned responses. To prove this claim, the authors analyze two unpublished fMRI pavlovian learning tasks in which participants learn to associate and then forget arbitrary visual cues with rectal distension or exteroceptive unpleasant signals (thermal pain in study 1, auditory cues in study 2). While several differences exist in the extinction phase between the two datasets (extinction phase is separated into two days in study two and visceral and exteroceptive unconditioned stimuli are reinstated simultaneously), the two datasets are well matched regarding task structure in the acquisition and extinction phase, and stimuli timing and visceral stimulation protocol are virtually identical, making it reasonable to compare between the two studies. Importantly, the level of pain/unpleasantness of each modality was carefully matched in the two studies, such that first, the threshold of visceral pain was obtained and exteroceptive pain or distress was matched to have a perceived, self-reported similar intensity, allowing to rule out differences in pain/unpleasantness between the modalities as source of differences in behavior and brain activation. fMRI data was analyzed with a series of first and second level analysis using an a-priori selected regions of interest consisting of insular cortex, mid and anterior cingulate cortex, hippocampus and amygdala. The effectiveness of the acquisition, extinction and reinstatement is assessed using subjective reports and visual analogue scales, with no objective markers of the effectiveness of the different conditioning stages, which in my opinion constitutes an important limitation to this study.

Their claim of visceral signals having a special status with respect to exteroceptive signals seems to be backed up by their results. Behaviorally, they find a significant difference in reported valence change before and after conditioning for the conditioned stimuli for interoceptive than exteroceptive threats, present in the two studies i.e. arbitrary visual cues predicting painful rectal distension are reported as more negative than those predicting external distress. Furthermore, the self-reported association between the conditioned and unconditioned stimuli was more difficult to extinguish in study 1, although not for study 2, the reasons of this discrepancy being ascribed to the fact that in study two, extinction phase was longer and occurred 24 hours later, which seems a reasonable possibility. Neurally, authors report increased brain activation for visceral vs external unconditioned stimuli in anterior insular and dorsal anterior cingulate cortex in both studies, as well as increased activation for interoceptive vs exteroceptive conditioned stimuli in mid cingulate and posterior cingulate cortex.

Response:

Thank you for this overall positive assessment and your helpful remarks. We fully agree with you (and other referees who have also raised this issue) that suitable objective physiological markers, such as SCR, galvanic skin or startle eyeblink responses, would substantially add to the manuscript. As we had disclosed in the original version of the manuscript, we did in fact measure electrodermal activity. Given some technical difficulties, our data was incomplete, which was why we did not include SCR analyses in the first version. Given the indisputable importance of such data, however, and in the interest of full transparency and reporting, as per request by another reviewer who specifically suggested that we include at least a subset of available data, we now have complied with this and present SCR analyses for a subset of participants for both studies (see supplemental Figure 5). While only available for a

subset, the results support prioritized learning in response to visceral pain-predictive cues, especially evident in study 2. In addition, we have amended the discussion to more explicitly and critically address this issue as a limitation and important future direction.

Please also allow us to comment on reaction time data. We would like to clarify that we did not obtain reaction time data in either study. Reaction time is not an established measure of associative fear learning neither in the broader fear conditioning literature (reviewed in Lonsdorf et al., 2017) nor in the pain-related fear conditioning literature (reviewed in Meulders, 2020), since the inclusion of reaction time tasks would not only introduce movement but also cognitive processes (e.g., decision-making, attention) that could interfere with classical conditioning processes to CS. We, and others, have however used reaction time assessment as a measure implemented *after* conditioning in order to assess the consequences of conditioning on attentional or perceptual decision-making processes or in avoidance conditioning (see for example, Labrenz et al., 2020; Zaman et al., 2017; Vervliet et al., 2017).

Original comment:

Overall, the study was carefully designed and executed and the claim goes in line with the obtained results. Regarding the novelty of the results, I have the impression that these results are well expected, based on previous results from the same team (REF 4), or more classical references (REF 5), and as such, are more confirmatory than mechanistic and provide no fundamental insight on the nature of the difference between visceral and somatic stimuli, the computational processes involved, or the recruitment of different brain regions. Furthermore I was expecting a more in depth discussion of the special status of visceral signals, which at several points seems redundant or tautological → e.g. lines 502-503 'Due to its interoceptive nature, visceral pain appears to be particularly threatening' lines 555-556 'the unique salience of visceral pain contributes to preferential interoceptive learning'.

What is the uniqueness of visceral signals that makes them more salient? Is it the salience of visceral signals that makes them special or rather the different anatomical pathways with respect to somatic signals? See Eickhoff 2008 for a classical fmri study on visceral vs somatic fmri activations and Azzalini 2019 for a list of references and a discussion regarding the special status of visceral signals.

References

Eickhoff, S. B., Lotze, M., Wietek, B., Amunts, K., Enck, P., & Zilles, K. (2006). Segregation of visceral and somatosensory afferents: an fMRI and cytoarchitectonic mapping study. *Neuroimage*, 31(3), 1004-1014.

Azzalini, D., Rebollo, I., & Tallon-Baudry, C. (2019). Visceral signals shape brain dynamics and cognition. *Trends in cognitive sciences*.

Gramsch, C., Kattoor, J., Icenhour, A., Forsting, M., Schedlowski, M., Gizewski, E. R., & Elsenbruch, S. (2014). Learning pain-related fear: neural mechanisms mediating rapid differential conditioning, extinction and reinstatement processes in human visceral pain. *Neurobiology of learning and memory*, 116, 36-45.

Response:

We appreciate the positive assessment of our experimental approaches. Regarding novelty, we have carefully revised the manuscript introduction and discussion to improve clarity regarding novel aspects. It is correct that our findings regarding visceral versus somatic pain processing are not entirely novel but rather confirm existing knowledge, which – in light of calls for replication – should in our view not devalue our manuscript. In fact, we have added a notable new publication (van Oudenhove et al., 2020) that focusses on common and distinct processing of visceral versus somatic pain stimuli using highly sophisticated experimental tools. Please also note that the comparison to an aversive auditory stimulus matched to unpleasantness is a novel addition. The most important point we would like to make is that while we included these analyses (herein: of visceral versus somatic and auditory US) to provide a comprehensive set of results, our primary focus is on conditioned cues (CS). This is why (in the interest of overall length of the manuscript and conciseness) we decided not to further expand on our present discussion of visceral signals themselves, but rather focus on modality-specific CS, i.e., conditioned expectations and underlying neural mechanisms. To the best of our knowledge, the CS data complement and extend existing studies in the field, and are particularly novel for the extinction and reinstatement-test phases.

Original comment:

My major concern with the paper is regarding the brain figures, which are not able to convey the location of the activation accurately and could potentially greatly confuse the reader (as in my case), raising concerns about the accuracy of the anatomical labeling. Too name a few of this confusing labelings;

-Figure 2A- dACC activation seems to occur in the ventricle. Left AI seems to occur in the brainstem

-2B Left insula activation occurs in the putamen and not in the insula

-Figure 4 Is particularly confusing, hippocampus in panel D and F seems to be in completely different location.

dAcc in panel G in the thalamus. Blob in the left part of panel G is unlabelled.

May I suggest a more standard coronal/sagittal/axial selection of slices, indicating the MNI coordinates and the axes? Or at least the same view for the different contrast?

Response:

This critical point is well-taken, and as suggested we have replaced all Figures in both the main manuscript and supplement with more standard 2D illustrations to improve clarity.

Original comment:

A second point is the used predefined ROIs. While the selection of ROIs seems justified, several relevant regions are left outside the analysis. Why not including Basal ganglia or secondary somatosensory cortices, which play an important role in either reinforcement learning or visceral pain, as from previous research from the group? Gramsch 2014. Additionally, might I suggest reporting whole brain uncorrected results in the supplementary results, and uploading the unthresholded t maps to neurovault.org, as well as the behavioral data to open science framework? This would increase the reproducibility and trust of the results.

Response:

These are also very valid points. As suggested, we now provide whole-brain uncorrected results for all contrasts. Tables within the main manuscript have been amended; additional Tables have been added to the supplement, and additional figures have been included in the supplement. Moreover, we have provided all unthresholded t-maps to neurovault.org, as requested (<https://neurovault.org/collections/GPPGVZAT>).

Original comment:

Finally, if exteroceptive and interoceptive stimuli were matched in unpleasantness, how come there are significantly greater reported post-acquisition US visceral unpleasantness ratings compared US somatic or auditory? Lines 386-389 are particularly confusing. Furthermore, while authors report observing no differences when using the self-reported unpleasantness data in their model, they do not provide the data while in fact the final, reported models of the paper should definitely include this important covariate.

Response:

As detailed in the method section, matching procedures (to intensity in study 1; to unpleasantness in study 2) were accomplished prior to conditioning. In other words, at baseline (prior to repeated US presentations during acquisition), US were successfully matched. After acquisition, US were no longer "matched", i.e., they differed in perceived intensity and unpleasantness, respectively, consistent with our previous observations (e.g., Koenen et al., 2017; Benson et al., 2019). These findings indicate possible differences in habituation, as we elaborated upon in Koenen et al., 2017, supporting the need to consider these subjective measures in analyses of BOLD responses. Therefore, we fully agree that it is necessary to include these data, and we now provide results with and without covariates in the main manuscript and supplemental files. Importantly, inclusion of covariates does not appreciably change results. We have also noted in the revised discussion that these results may indicate modality-specific differences in habituation, but given the focus of the manuscript on CS (rather than US) and word limit, we respectfully refrain from further more detailed discussion of habituation processes.

Comments from Reviewer #2:

Original comment - Review Overview:

Apologies for my late review, as this article came to my desk during the corona crisis. This is a beautifully conceived and executed paper on an extremely timely topic. In brief, the authors investigated the behavioural and neural effects of interoceptive versus exteroceptive condition stimuli and responses. Using a well-established protocol developed by this lab, they compare the influence of interoceptive (rectal distension) vs exteroceptive (thermal) pain stimuli on subjective valence ratings and fMRI responses during acquisition, extinction, and recall. The study employs a well-controlled and elegant experimental design with numerous positive controls and validation checks. Overall, the results are robust, meaningful, and coherent with what can be expected given previous work in this area. Additionally, I think these results speak to a larger debate in the predictive processing community regarding the role of interoceptive signalling in hierarchical precision weighted inference. I think this paper will have a large impact on the field and generate numerous follow-on behavioral, neural, and clinical studies. My comments consider primarily improving the transparency and reproducibility of these excellent findings, clarifying a few theoretical points, and opening up the discussion to those outside the pain conditioning literature.

Response:

Thank you. We greatly appreciate your time, interest in our work, and the positive and constructive comments that motivate us not only to further improve this manuscript but to continue and broaden our research efforts in this fascinating area of work not only within but also outside the pain conditioning realm.

Original comment:

1. This is an excellent study. The primary results show that interoceptive conditioned stimuli evoke a greater level of negative valence than matched exteroceptive control stimuli, and that this effect is particularly pronounced during reactivation in the extinction phase. Evolutionarily speaking, this finding makes a lot of sense: errors in the interoceptive modality are almost always life threatening and warrant extreme, 'one-shot' learning. As a simple example, consider how a single case of food poisoning is generally enough to evoke fear and avoidance from that food, and even slightly similar foods, for a lifetime. I think these results speak beautifully to some of these issues, so I just want to congratulate the authors.

Response:

Again, thank you very much for your appreciation and positive assessment of our work. We could not agree more with your evolutionary perspective, and have incorporated some of these consideration into our revised manuscript (see revised introduction and discussion).

Original comment:

Along these lines, have the authors looked at the history of interoceptive conditioning literature carried out by the soviet psychologists? I see their experiment very much as a modern replication of these classical results, which is a great thing. In particular, as reviewed by Rezlan (1972, ref#9), the soviet psychologists claimed to find that interoceptive traces were more rapid to acquire, more strongly resisted extinction, and could be reactivated more easily than exteroceptive traces. I think it would be lovely if the authors could contextualize their results in light of these classical findings and note that it detracts absolutely nothing from the novelty here, which is substantive.

Response:

We greatly appreciate these suggestions, and have amended the paper accordingly. Indeed, some conditioning work by soviet psychologists remains underappreciated, admittedly also by us, a shortcoming which we hope to improve in our revised version. Thank you very much for encouraging us to further explore the history and incorporate classical findings.

Original comment:

I really do respect that the authors are very circumspect in their claims and theorizing. Almost all of the results are discussed using relatively theory neutral terms like “salience”, couched in the basic language of classical conditioning. In this case, I want to ask the authors to speculate a bit more about these results, in light of the rapidly emerging interoceptive predictive processing literature. A major debate there concerns the “precision” or confidence assigned to visceral signals (both ascending errors and descending predictions). While many have assumed visceral signals are imprecise (ref #), the classical results mentioned in the previous points suggest that deviations on the internal channel seem to be assigned a much higher weight than those on the exteroceptive channel, perhaps as an evolved survival mechanism. This actually speaks to the psychological definition of salience itself, as a weighting of information by whatever yields the best evidence for one’s long-term survival. While I don’t expect the authors to go into this level of detail, it would greatly expand the impact of the article if they could connect to this more general literature on interoception and predictive processing. Obviously, without a distinct computational model at hand the authors can at best speculate about the mechanisms at play in their results, but I do believe at least a little speculation is warranted here. See refs 3-8, which could be worked into the discussion at the author’s discretion.

Response:

We absolutely agree, and followed these valuable suggestions. We incorporated additional references (and would especially like to draw this reviewer’s attention to van Oudenhove et al., 2020) and expanded the discussion section. Given the primary focus on CS and the already long discussion, please accept our apologies for not going into the level of detail that the topic actually deserves. We strive to do so in future work, fully convinced that this deserves dedicated attention.

Original comment:

The authors analyse only subjective valence ratings, which are certainly interesting and at least partially independent from their control of overall pain intensity. However, given the behavioral paradigm I was left wondering why no reaction time data are analyzed. If the interoceptive association is learned more readily, could this not be seen in the RTs themselves? I welcome the subjective data analysis, but it could be strongly complemented by a more behavioral measure of learning, if one is available.

Response:

Point-well taken. As also detailed in our response to a comment from another referee (reviewer 1, please see first comment and response) in the same direction, we have amended the manuscript to include analyses of the SCR data (see Figure S6) we had available, and amended the discussion. We did not measure RTs in either study.

Original comment:

I feel like the one weakness of the article is in the neuroimaging results. I am generally not a fan of ROI analyses unless the ROIs have been specifically pre-registered prior to any data collection or analysis, as it is simply too easy to inflate results using this approach. I am not saying the authors did this at all, and I do think their ROIs are sensible given the domain, but it is something I consistently flag as I think we need to be moving away from non-preregistered ROI analyses, generally speaking. With that said, the whole brain results as plotted look a bit odd; many of the ‘insula’ or ‘cingulate’ activations look focused in the white matter or CSF. This could potentially imply an issue with nuisance covariance. From my count the authors utilize more than 5 ROIs and 5 contrasts, so the inflation of the family wise false positive error rate could be greatly inflated, even with the FWE peak level correction applied. I think the results should ideally be supplemented with a whole brain analysis.

Response:

We completely agree, and have as suggested made substantial changes to the illustration of the results (all figures revised) and included whole-brain results in all tables reporting BOLD data (both within main manuscript and supplement), and also added new whole-brain illustrations in the supplement. Please note that while we unfortunately did not pre-register ROIs, our work is funded within a research

consortium of the German Research Foundation on extinction learning, and our grant proposal detailed the ROIs that were actually the focus of multiple subprojects by different independent research groups using different approaches to assess neural mechanisms and clinical implications of extinction learning.

Original comment:

Following on from the above, my primary worry here would be dealing with white noise in the fMRI signal arising from cardiorespiratory fluctuation. Do the authors have any pulse oximetry or respiration belt data? I could easily imagine that the more salient interoception condition might cause “gasping” or increased respiration and heart rate. Ideally, I would like to see a RETROICOR or similar control analysis ensuring that the activation results are not vascular artefacts, which are particularly amplified in the insula and anterior-mid cingulate. At the very least, some analyses of physiological summary statistics would partially assuage these concerns.

Response:

Thank you for these critical yet constructive considerations. While we unfortunately did not measure pulse oximetry or respiration and hence are unable to perform a RETROICOR control analysis, we now provide an alternative supplemental analysis as a proxy: As specified in our original manuscript, study 1 included a habituation phase prior to acquisition, which we did not analyse for the purposes of the present manuscript. In an attempt to explore possible pain modality-specific movements (and assuming that gasping responses would result in rhythmical movement right after pain stimulus onset), we analysed head movement parameters (linear and degree movement) measured during this habituation phase which consisted of five visceral and five somatic pain stimuli delivered in pseudorandomized order. Results are provided in novel supplementary figure M1, and give no indication of a rhythmical response to the onset of either pain stimulus that would be suggestive of a gasping response. The method section of the main manuscript has been critically revised to point out this potential problem and incorporate the novel supplemental analysis and figure. In the discussion, we also now draw explicit attention to this potential limitation.

Original comment:

Additionally, the authors use a sub 2s TR, but applied the default AR(1) technique for whitening. For faster TRs in this range, my understanding is that the SPM authors now recommend the FAST technique, which can better handle aliasing effects that are pronounced at faster acquisition speeds. I think a supplementary control analysis is definitely called for.

Response:

Thank you for raising awareness for the very important matter of choosing the right model for whitening the data to accounting for temporal correlations, which is of high relevance in rapid imaging techniques. However, we believe that there is a misunderstanding and we are happy to clarify: For our experiments, we chose a TR of 2300ms for our functional data acquisition; 1900ms was used only for structural data acquisition). Therefore, the AR(1) model approach seemed suitable (Corbin et al., 2018, *Hum Brain Mapp.*; Bollmann et al., 2018, *Neuroimage*). We have adapted the respective methods section to clarify.

Original comment:

Openness. This is an excellent article, using a robust experimental design and I applaud the authors for accurately reporting effect sizes alongside statistical tests. However, as a signatory of the COBIDAS guidelines for reproducibility in neuroimaging, I must ask them to go further. Specifically:

Nowhere is there any mention of data and/or code availability. The authors should upload their anonymized behavioral and neuroimaging data to a repository of their choice, and also make available all analysis and paradigm scripts. This is essential for the reproducibility of their important findings. If they are unable to do so, e.g. due to ethical constraints or similar, then it should be explicitly stated within the article why they have chosen not to share these materials, and what steps future authors should take to obtain them. I understand that in many cases it may not be possible to share raw or

subject level biomedical imaging data: in this case, uploading the group level summary t-contrast maps to Neurovault is a sufficient minimum.

Response:

We completely agree. As described in the methods section, analyses were conducted with default settings as predefined by the respective software programs (SPSS for behavioural and SPM for BOLD data analyses). However, we have added a data availability statement and will provide any code and/or analysis script, if requested by readers. Moreover, we have uploaded all unthresholded t-maps in Neurovault as suggested.

Original comment:

Robustness in data visualization. The authors use bar plots in several places to illustrate their behavioral and neuroimaging results, but these can be highly misleading, and obscure the underlying nature of the data. In the interest of transparency and rigor, I encourage the authors to use a more robust plotting approach such as Rainclouds or Beeswarms. See refs # and # for more information and examples.

Response:

Done as suggested. Thank you for providing the very helpful tools in order to produce more transparent and rigorous data plots. Please see revised visualization of data in Figures 3 and 4 (main manuscript) and Figure S6 (supplement) as Rainclouds.

MINOR

Original comment:

1. Page 6, line 159 – here and elsewhere in the manuscript the authors use the phrase “fMRI was accomplished”, but this is incorrect grammar, and an unusual wording. A more correct formulation would be “fMRI was applied to assess..”.

Response:

Thank you. Rephrased as suggested.

Original comment:

2. Page 15, line 464-465 ‘the interaction approached significance’. Frequentist tests do not approach, flirt with, or comingle with significance. There are either significant or not, based on the a priori alpha criterion. I don’t believe the authors rest any interpretive weight on these results (if they do so, then they should not), but this phrasing should be revised and generally avoided. If you want a continuous measure of evidence, use a Bayesian or likelihood-based approach.

Response:

Point well-taken. We have rephrased (see line 467-469) within the results section, and indeed do not base interpretive weight on this result.

Original comment:

Finally, I would like to thank the authors for their patience, and their hard work on this excellent manuscript. Following suitable revision I look eagerly forward to its publication.

Response:

Again, thank you very much for your insightful comments and helpful remarks that unequivocally helped us to further improve the quality of our manuscript.

Comments from Reviewer #3:

Original comment:

In this study, the authors compared acquisition, extinction, and reinstatement of threat in two fMRI studies using visceral vs. non-visceral USs. They found greater responding to visceral USs, and CSs that predicted visceral USs in acquisition and reinstatement in regions of the fear/salience network. This is an interesting study with an adequate sample size and a built-in replication. For the most part, they used robust data collection and analysis techniques (Exceptions discussed below). Given the uniqueness of the study procedures and the rigor of the methodology, I believe that with substantial revisions this manuscript would be of interest to the readers of Communications Biology.

Response:

Thank you for your overall positive assessment of our work. We hope that the revised version is further improved owing to the knowledgeable and constructive feedback from this reviewer and the two other referees.

Original comment:

My primary concern with this manuscript is that the USs were perceived differently after acquisition, which makes it difficult to determine whether their results were due to US intensity or US modality (See 408-410 and 419-421). The authors mention that they have included US rating as a covariate in some of the analyses, but that the data are not included. I would recommend including these data to allow the readers to determine themselves whether the US intensity impacted the findings.

Response:

We completely agree, and now provide full results of all analyses with the respective covariates (ROI analyses, and whole-brain analyses). Please see supplementary tables S6-S9.

Original comment:

Given that the US intensity was matched at the start of the experiment, a likely candidate for the differences in post conditioning US ratings may be habituation. The authors should discuss the available literature addressing differences in habituation among visceral vs. non-visceral stimuli.

Response:

This point is very well-taken. As suggested, we have included a brief note of possible modality-specific differences in habituation processes in our revised discussion. Given the focus of our manuscript on CS rather than US and the already lengthy discussion, we decided not to discuss this in greater depth.

Original comment:

The authors included a set of correlations between CS-related activation differences across modalities and US-related differences across modalities (See 429-434). This seems like double-dipping, especially considering the above concerns about post conditioning differences in US intensity ratings.

Response:

We appreciate this consideration, and have moved the exploratory correlational analyses to the supplemental section. After careful consideration, we believe that they do constitute an interesting addition, which is why we decided not to omit them entirely.

Original comment:

Throughout the reinstatement section, the authors report the presence/absence of effects in one group vs. another the group x variable interaction (e.g. 460-464). This is not a statistically valid approach. The authors should conduct an omnibus analysis with group as a factor to truly test whether the presence of an effect in one group over another reflects a true significant interaction.

Response:

This important point that was also raised also by reviewer #2. We have changed this accordingly, please see revised methods and results.

Minor issues

Original comment:

The cutaway figures showing differential activation are distracting and difficult to interpret. With the clusters of activation floating in space, it is difficult to precisely localize their anatomy. If the authors choose to include this figure, they should also include a montage of axial/sagittal/coronal views of highlighted results so that the anatomy can be precisely inspected.

Response:

Point well-taken. All figures have been changed.

Original comment:

On lines 228-231, the authors mention that the total duration of the US was matched across modalities. This statement is confusing. Were the individual durations variable? How long was each US presented?

Response:

We apologize for the lack of clarity. We have revised the respective method sections within the main manuscript and supplement (Methods 1) accordingly.

Original comment:

For study 2, would make sense to include a supplemental analysis with same number of extinction trials as in study 1 for comparison (See lines 445-449).

Response:

Thank you for this valid and relevant suggestion. We have added this analysis (see Table S4). Since results show no differences even after a “short” extinction (here: mid-extinction time point for study 2), we have also revised the discussion of these findings, which now more strongly point to a role of consolidation and less clearly to the role of number of extinction trials. Clearly, more studies are needed to clarify further, which we emphasize in our discussion of these findings.

Original comment:

If it is possible to include a portion of the SCR data as a supplement, this would be a welcome addition.

Response:

Done as suggested, please see additional analyses of a portion of SCR data, now provided as supplementary figure S6.

REVIEWERS' COMMENTS:

Reviewer #2 (Remarks to the Author):

Thanks to the authors for their thorough work on the revision. All of my comments have been adequately assessed. This paper is an excellent contribution to the literature.

Reviewer #3 (Remarks to the Author):

My Comments have been fully addressed. No further changes are needed.